# Homeostatic control of START through negative feedback between Cln3-Cdk1 and Rim15/Greatwall kinase in budding yeast

Nicolas Talarek, Elisabeth Gueydon, Etienne Schwob*

IGMM, CNRS, University of Montpellier, Montpellier, France

**Abstract** How cells coordinate growth and division is key for size homeostasis. Phosphorylation by G1-CDK of Whi5/Rb inhibitors of SBF/E2F transcription factors triggers irreversible S-phase entry in yeast and metazoans, but why this occurs at a given cell size is not fully understood. We show that the yeast Rim15-Igo1,2 pathway, orthologous to Gwl-Arpp19/ENSA, is up-regulated in early G1 and helps promoting START by preventing PP2A$^{Cdc55}$ to dephosphorylate Whi5. *RIM15* overexpression lowers cell size while *IGO1,2* deletion delays START in cells with low CDK activity. Deletion of *WHI5*, *CDC55* and ectopic *CLN2* expression suppress the START delay of *igo1,2Δ* cells. Rim15 activity increases after cells switch from fermentation to respiration, where Igo1,2 contribute to chromosome maintenance. Interestingly Cln3-Cdk1 also inhibits Rim15 activity, which enables homeostatic control of Whi5 phosphorylation and cell cycle entry. We propose that Rim15/Gwl regulation of PP2A plays a hitherto unappreciated role in cell size homeostasis during metabolic rewiring of the cell cycle.

*For correspondence: etienne. schwob@igmm.cnrs.frschwob@ igmm.cnrs.fr

**Competing interests:** The authors declare that no competing interests exist.

## Introduction

The size and shape of cells are main determinants of their function. Despite large differences in cell sizes within and between species, cells of a given cell type usually adopt a homogenous size suited to their nutritional environment, a property called cell size homeostasis. Even after stress-induced cell size variations, cells return to their original size within a few cell divisions (*Fantes, 1977*). Such size correction mechanisms imply that cells have means to sense their size and to change their division rate accordingly (*Ginzberg et al., 2015*; *Jorgensen and Tyers, 2004*). Budding yeast and mammalian cells coordinate cell growth and cell division mostly in G1 at a point called 'START' or 'Restriction point', respectively, after which they become committed to undergo cell division (*Johnston et al., 1977*; *Killander and Zetterberg, 1965*). In fission yeast this size control operates mainly in G2 (*Nurse, 1975*). In *S. cerevisiae*, a large body of evidence indicates that the earliest G1 cyclin, Cln3, fulfills the criteria required for a size sensor: it has a short half life and its abundance scales with protein synthesis rates; increased Cln3 dosage makes cells smaller, and conversely low Cln3 makes cells larger. Cln3-Cdk1 promotes START by phosphorylating the retinoblastoma (Rb) ortholog Whi5, thus activating the SBF transcription factor responsible for expression of the late G1 cyclins Cln1,2 and cell cycle entry (*Costanzo et al., 2004*; *Cross, 1988*; *de Bruin et al., 2004*; *Jorgensen et al., 2002*; *Nash et al., 1988*). However, what precisely sets the right size for START is unclear because Cln3 concentration seems to remain constant during G1. Several models have been proposed to explain how Cln3 could trigger START at a specific cell size: (i) rise in nuclear Cln3 (*Jorgensen et al., 2007*), (ii) Cln3 sequestration in the endoplasmic reticulum (*Vergés et al., 2007*; *Yahya et al., 2014*), (iii) titration of Cln3 against fixed number of SBF sites (*Wang et al., 2009*) or (iv)

dilution of the Whi5 inhibitor (*Schmoller et al., 2015*). However each of these models has downsides and none fully explains how the critical cell size required for START is modulated by nutrients (*Jorgensen and Tyers, 2004*; *Schmoller and Skotheim, 2015*). In particular, it is not known why yeast cells grown on poor nutrients, which have low Cln3 levels and little biosynthetic capacity, actually pass START at a smaller cell size.

START occurs when Cln3-Cdk1 phosphorylates Whi5 and Stb1, changing them from repressors to activators of SBF, which controls the transcription of ~200 genes important for G1 progression and DNA synthesis, including the *CLN1* and *CLN2* cyclins (*Costanzo et al., 2004*; *de Bruin et al., 2004*; *Wang et al., 2009*). Cln1,2 accumulation leads to further SBF activation through positive feedback, making the G1/S transition irreversible (*Charvin et al., 2010*; *Cross and Tinkelenberg, 1991*; *Dirick and Nasmyth, 1991*; *Skotheim et al., 2008*). Full Cln1,2-Cdk1 activity then triggers degradation of Sic1, an inhibitor of S-phase CDKs (Clb5,6-Cdk1), which leads to the initiation of DNA synthesis (*Nash et al., 2001*; *Schwob et al., 1994*; *Schwob and Nasmyth, 1993*). A similar pathway operates in mammalian cells with cyclin D-Cdk4 phosphorylating retinoblastoma family members (Rb, p107, p130) to relieve the inhibition of E2F, leading to synthesis of cyclin E, degradation of the p27[Kip1] CDK inhibitor and initiation of DNA synthesis (*Bertoli et al., 2013*). The G2/M transition is also triggered by positive feedback, with M-CDK promoting degradation of its inhibitor Wee1 and stabilization of the counteracting Cdc25 phosphatase (*Pomerening et al., 2003*). The mechanisms leading to the initial activation of these auto-regulatory, positive feedback loops controlling G1/S and G2/M are less well understood (*Kishimoto, 2015*).

The phosphorylation state of target proteins depends not only on kinase activity but also on the balance with counteracting protein phosphatases (*Uhlmann et al., 2011*). Hence inhibiting phosphatases is another way to promote CDK-driven cell cycle transitions (**Figure 1A**). For example Cdc14, the main phosphatase counteracting M-CDK phosphorylation in budding yeast, is sequestered in the nucleolus away from its mitotic substrates and only released during anaphase when M-CDK sites need to be dephosphorylated for M exit (*Visintin et al., 1998*). Another major phosphatase, PP2A, was considered constitutively active until it was found that the Greatwall/Mastl kinase transiently inhibits PP2A-B55 via phosphorylation of the endosulfines ENSA and Arpp19 (*Gharbi-Ayachi et al., 2010*; *Mochida et al., 2010*). This inhibition of PP2A is required for robust mitotic entry and progression. Recent data and mathematical modeling suggest that inhibition of PP2A by the Gwl pathway is key for irreversible, switch-like mitotic entry in *Xenopus* egg extracts (*Mochida et al., 2016*). The Gwl signalling pathway is strikingly conserved on the biochemical level in yeast, with Rim15 kinase phosphorylating Igo1 and Igo2, which then inhibit PP2A[Cdc55] (*Bontron et al., 2013*; *Juanes et al., 2013*; *Talarek et al., 2010*). The function seems different however as *S. cerevisiae* Rim15-Igo1,2 play key roles in quiescence entry and gametogenesis, but only a minor role in mitosis upon cellular stress (*Juanes et al., 2013*; *Sarkar et al., 2014*). Interestingly, both Rim15 and its fission yeast ortholog Ppk18 are regulated by TORC1, thereby modulating cell cycle progression and cell size in response to nutrients (*Chica et al., 2016*; *Martín et al., 2017*; *Moreno-Torres et al., 2015*; *Pedruzzi et al., 2003*). Recently, it was found that human cells depleted of ENSA have a longer S phase due to decreased levels of Treslin/TICRR, an essential replication initiation protein, indicating that Gwl does more than only regulating mitosis in mammals (*Charrasse et al., 2017*).

By reinvestigating the role of the Rim15-Igo1,2 pathway during the vegetative cell cycle, we found that it promotes START by facilitating Whi5 phosphorylation in conditions of low Cln3-Cdk1 activity. Rim15 is activated when cells are grown on poor nutrients, and inhibits PP2A[Cdc55] that would otherwise maintain Whi5 in its dephosphorylated, START-repressive state. We show that Rim15 activity is regulated by Cln3-Cdk1 levels, so that the two kinases homeostatically control the size at which cells enter the cell cycle. We propose that the Rim15-Igo1,2-PP2A pathway modulating Whi5 phosphorylation is the long-sought component that fine tunes START in response to nutrient conditions.

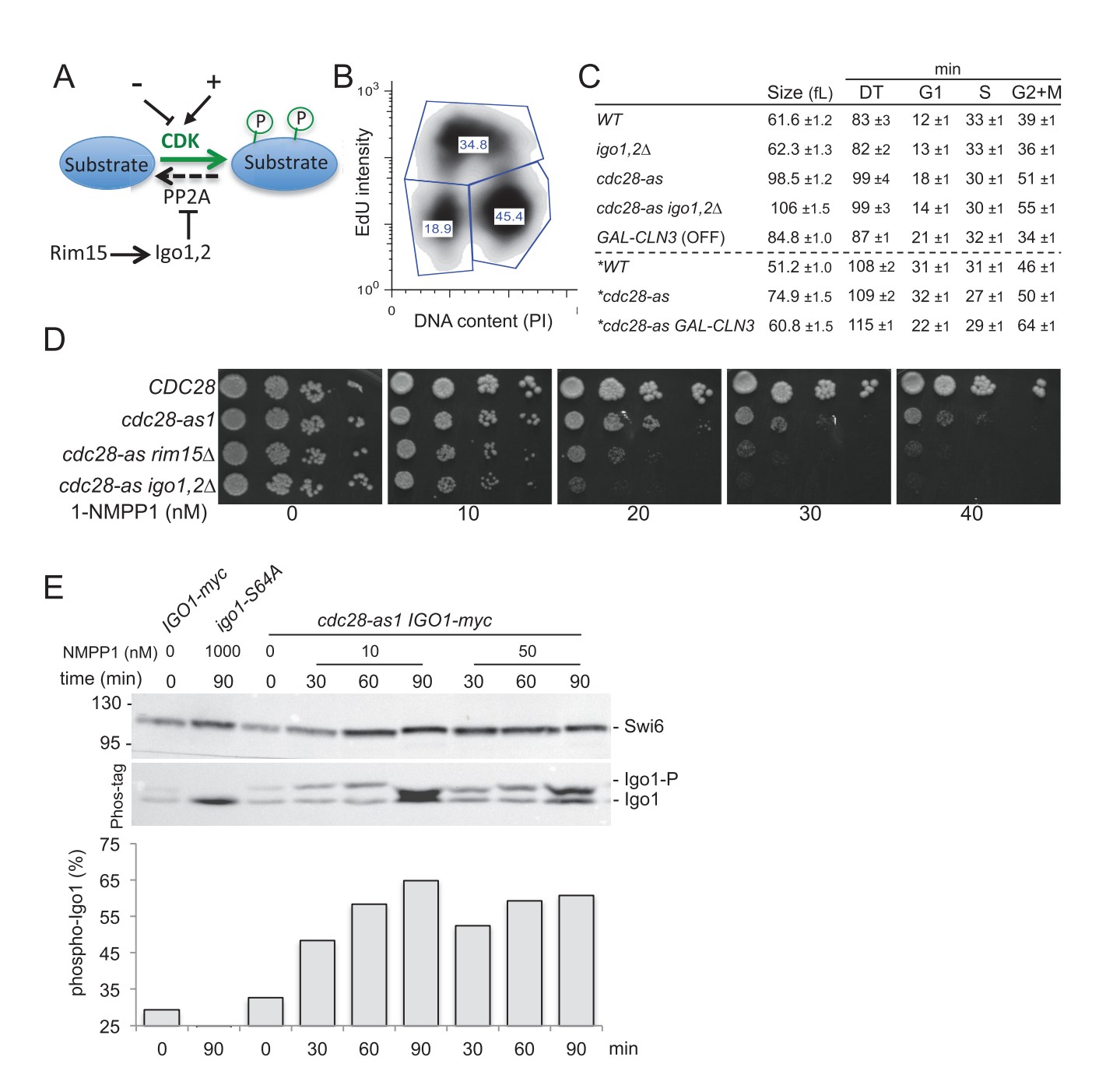

**Figure 1.** The Rim15 pathway is activated and becomes essential when Cdc28 activity is low. (**A**) The Rim15 pathway can potentiate CDK-dependent phosphorylation of target proteins by inhibiting the CDK-counteracting PP2A phosphatase. (**B**) Bivariate EdU/PI FACS profile of asynchronous *WT* cells (E3087) pulsed for 10 min with 25 μM EdU. The proportion of G1 (lower left polygon), S phase (upper polygon) and G2+M cells (lower right polygon) is indicated. (**C**) Median cell volume (in fL), doubling time (DT in min) and the duration of the G1, S and G2+M phases (in min) is indicated for each strain grown in SCD or SCRaf+Gal (asterisk) medium at 30°C in the absence of 1-NMPP1 (E3087, E4259, E4479, E4458, E5493, E5492). Measures were done in triplicate and represented as Mean ± SEM. (**D**) Cells of the indicated genotypes (E3087, E4479, E4471, E4458) were spotted in fivefold serial dilution on YPD plates containing increasing concentrations of 1-NMPP1. Rim15 and Igo1,2 become essential for viability in *cdc28-as1* cells exposed to more than 20 nM 1-NMPP1. (**E**) Analysis of Rim15 activity by monitoring the level of Igo1 phosphorylation on Phos-tag gels. *IGO1-myc, igo1-S64A-myc and cdc28-as1 IGO1-myc* cells (E4974, E4975 and E4996, respectively) were exposed for 0, 30, 60 or 90 min to 10 or 50 nM 1-NMPP1 and Igo1 phosphorylation was examined by western blotting of Phos-tag gels. Quantification of the fraction of Igo1-S64 phosphorylation is shown below. Swi6, loading control.
*Figure 1 continued on next page*

*Figure 1 continued*

The following figure supplements are available for figure 1:

**Figure supplement 1.** Hypersensitivity of *cdc28-as1 igo1,2Δ* cells is due to G1, not G2, defect.

**Figure supplement 2.** The Rim15-Igo1,2 pathway controls cell size.

## Results

### Lowering CDK activity reveals a cell cycle function for the Rim15-Igo1,2 pathway

Gwl/Mastl plays a role in mitotic entry and progression in *Drosophila*, *Xenopus* and mammals, yet ablation of the orthologous Rim15 kinase or Igo1-Igo2 endosulfines has little or no effect on cell cycle progression in yeast, at least in rich medium and unstressed conditions (*Juanes et al., 2013*; *Rossio et al., 2014*; *Thai et al., 2017*). We recently devised a flow cytometry method based on pulse incorporation of the thymidine analogue ethynyl-deoxyuridine (EdU) in yeast to precisely determine the fraction of cells in G1, S and G2+M phases of the cell cycle (*Figure 1B*) (*Talarek et al., 2015*). The duration of each phase can be calculated by multiplying their fraction by the population doubling time. Using this approach we found no significant difference between *WT* and *igo1Δ igo2Δ* cells neither in the duration of G1, S and G2+M, nor in median cell size or doubling times (*Figure 1C*), consistent with earlier studies. This absence of phenotype might stem from CDK activation being strong enough, in regular laboratory conditions, to drive cell cycle transitions without the need to inhibit the CDK-counteracting PP2A phosphatases (*Figure 1A*). Since major cell cycle transitions are often controlled by multisite phosphorylation of key substrates (*Holt et al., 2009*; *Nash et al., 2001*), we reasoned that the Rim15/Gwl phosphatase-regulating pathway might come into play only when CDK activity decreases below a threshold where unrestrained phosphatase activity would prevent CDKs to fully phosphorylate these key substrates.

To test this idea, we used the *cdc28-as1* Shokat allele of Cdk1 whose activity can be inhibited specifically and gradually with the bulky ATP analogue 1-NMPP1 (*Bishop et al., 2000*). While *CDC28* cells were insensitive to all tested doses of 1-NMPP1 (up to 1000 nM), *cdc28-as1* cells showed severe growth defects on YEPD plates containing 40 nM 1-NMPP1 or more (*Figure 1D*). Interestingly, deletion of *RIM15* or *IGO1* and *IGO2* rendered *cdc28-as1* cells hypersensitive to the drug, with growth defects at 20 nM 1-NMPP1. The 1-NMPP1 hypersensitivity of *cdc28-as1 igo1,2Δ* cells is likely due to the lack of PP2A inhibition since *cdc28-as1 igo2Δ* cells containing a non-inhibitory Igo1-S64A protein (that cannot be phosphorylated by Rim15) were similarly hypersensitive to the drug (*Figure 1—figure supplement 1A*). Earlier studies showed that *rim15Δ* and *igo1Δ igo2Δ* mutants are sensitive to temperature stress and delay G2/M in a *SWE1*-dependent manner (*Juanes et al., 2013*). In contrast, the 1-NMPP1 hypersensitivity was not suppressed by *SWE1* deletion, suggesting that it is not due to delayed mitotic entry (*Figure 1—figure supplement 1B*). The fact that Rim15 and Igo1,2 become essential when Cdk1 activity is lowered could indicate that the pathway is hyperactive in these conditions. Indeed using Phos-tag gel analysis (*Kinoshita et al., 2006*) to monitor Igo1 phosphorylation, we found that the pool of Igo1 phosphorylated at Ser64 raised from 32% to 65% upon Cdk1 attenuation with only 10 nM 1-NMPP1 (*Figure 1E*). We conclude that decreasing Cdk1 activity causes an increase in Rim15 activity measured on Igo1.

We noticed that *cdc28-as1* cells were much larger (population median cell volume 98.5 fL) and had a longer doubling time (99 min) than *CDC28* cells (61.6 fL, 83 min), even in the absence of 1-NMPP1 (*Figure 1C*). This is likely due to the lower ATP binding and turnover of the *cdc28-as1* kinase (*Bishop et al., 2000*). Using bivariate EdU-PI FACS analysis coupled to doubling time we found that the G1 and G2+M phases were lengthened, indicating that both the G1/S and G2/M transitions are affected by the *cdc28-as1* mutation (*Figure 1C*). Addition of 1-NMPP1 mostly delayed mitosis, however (*Figure 1—figure supplement 1C*). Importantly, Cln3 overexpression suppressed the large size and the long G1 of *cdc28-as1* cells grown without 1-NMPP1, indicating that the *cdc28-as1* protein likely has defects in the interaction with, or activation by G1 cyclins, with consequences on size

control (*Figure 1C* and *Figure 1—figure supplement 1D*). Deletion of *IGO1* and *IGO2* further increased the median size and coefficient of variation of *cdc28-as1* cells grown in the absence of 1-NMPP1, without changing their doubling time (*Figure 1C*, *Figure 1—figure supplement 2*). Note that the apparent shorter G1 duration in *cdc28-as1 igo1,2Δ* cells stems from them being born at a larger size, not because they pass START earlier. Conversely expression of *RIM15* from the strong *ADH1* promoter lowered the size of wild type, *cdc28-as1* and *cln3Δ* cells, reinforcing the notion that the Rim15-Igo1,2 pathway is involved in size control (*Figure 1—figure supplement 2*). Most size regulation operating in G1 in budding yeast, our data suggest that the Rim15 pathway has a cryptic role in G1 control. This is consistent with studies showing synthetic lethality between *rim15Δ* and *cln2Δ* (*Fiedler et al., 2009*) and a function for Rim15 in quiescence entry as well as in early meiosis (*Sarkar et al., 2014*; *Talarek et al., 2010*).

## The Rim15 pathway is hyperactive in early G1 cells and repressed by Cln3-Cdk1

Juanes et al. concluded that Igo1 phosphorylation is constant during the cell cycle, but this was done after release from late G1 (α-factor) arrest (*Juanes et al., 2013*). To assess Rim15 activity throughout G1 we size-selected early G1 *IGO1-myc* cells by centrifugal elutriation, released them in SC-D medium and analyzed Igo1 phosphorylation during the cell cycle using Phos-tag western blotting. While total Igo1 levels were constant, Igo1 phosphorylation was highest in early G1 cells, decreased ~2.5 fold around the time of budding and then remained low for the remainder of the cell cycle (*Figure 2A* and *Figure 2—figure supplement 1A*). It is known that Igo1-S64 phosphorylation depends on Rim15 (*Bontron et al., 2013*; *Chica et al., 2016*). No shift was detected in the *igo1-S64A* mutant, indicating that phosphorylation is due to Rim15. We conclude that Rim15-dependent phosphorylation of Igo1-S64 is maximal in early G1, leading to a tighter inhibition of PP2A$^{Cdc55}$ before START of the cell cycle.

Since Igo1 phosphorylation decreased around the time of budding, we tested whether it might be regulated by G1 CDK activity. To this end, we used asynchronous cultures of a *cln1Δ cln2Δ GAL-CLN3 IGO1-myc* strain in which the sole source of G1 cyclins (Cln3 in this case) is under control of the *GAL1* promoter. Cells were grown in SC-RafGal, centrifuged and resuspended either in SC-RafGal (*CLN3* ON) or in SC-RafGlu (*CLN3* OFF), then aliquots were taken every 10 min for quantitation of Igo1-S64P on Phos-tag gels. *Figure 2B* shows that Igo1-S64 phosphorylation doubled within 20 min of *CLN3* repression, and conversely decreased twofold within 10 min of transfer to Gal medium (*CLN3* ON). Importantly, these changes in Igo1 phosphorylation occurred in asynchronous cells and much faster than any change in cell cycle distribution, ruling out the possibility that they are linked to cell cycle position. This result indicates that Rim15 activity responds rapidly to variations in Cln3 levels at any position of the cell cycle (*Figure 2—figure supplement 1C*). We conclude that Rim15 activity is highest in early G1 cells, and repressed when Cln3-Cdk1 activity rises at START. The idea that Cln3-Cdk1 represses Rim15 is consistent with Rim15 activity being highest in early G1 cells (*Figure 2A*) when Cln3 is lowest (*Thorburn et al., 2013*; *Zapata et al., 2014*).

## The Rim15-Igo1,2 pathway promotes START by inhibiting PP2A$^{Cdc55}$

What could be the function of Rim15 in early G1 cells? One possibility is that it inhibits PP2A, via Igo1,2 phosphorylation, to favor the phosphorylation of CDK substrates important for START or the initiation of DNA synthesis. Hence we compared the cell size at START, as well as the initiation and duration of S phase, after elutriation of *WT* and *igo1Δ igo2Δ* cells. To correct for any differences in cell synchrony between different strains and cultures, we scored the median cell volume of the population when it reached half-maximal budding (*Figure 2—figure supplement 1B*), as in other studies of cell size at START (*Jorgensen et al., 2002*; *Thorburn et al., 2013*). This size was only marginally increased, if at all, in *igo1,2Δ* cells compared to wild type (*Figure 3A* left), as expected from the unchanged cell size, doubling time and cell cycle phase durations measured on asynchronous populations. However in conditions of lowered CDK activity, using the *cdc28-as1* allele without 1-NMPP1 addition, we found that deletion of *IGO1,2* increased the cell size at budding by 30% (*Figure 3A* right). The increased START size of *cdc28-as1 igo1,2Δ* cells was confirmed on single cells expressing Whi5-sfGFP, which is nuclear from late M until START (*Figure 3—figure supplement 1A*). These data indicate that the Rim15 pathway promotes START, but only in conditions of decreased CDK

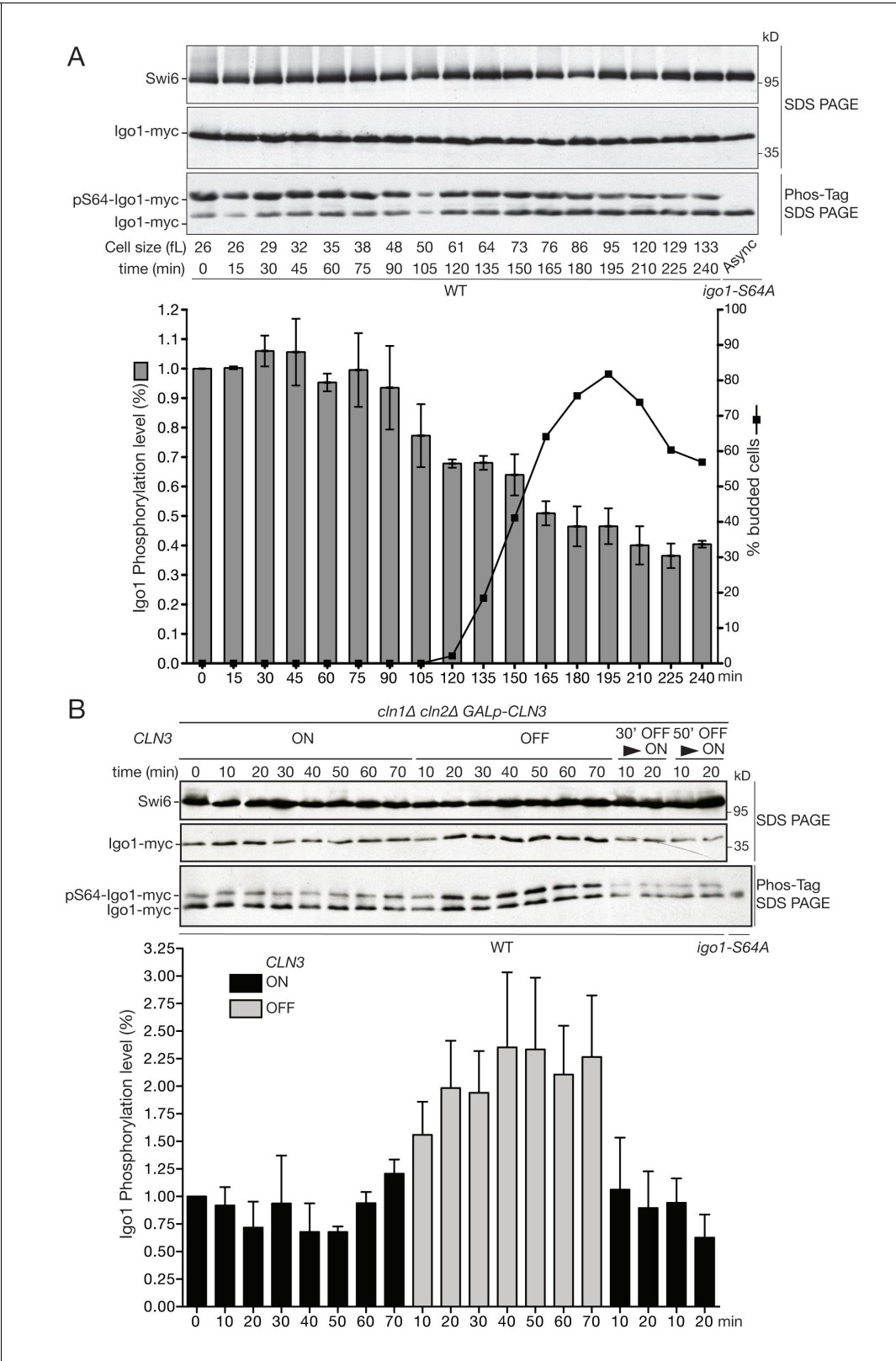

**Figure 2.** The Rim15 pathway is active in early G1 cells and negatively regulated by Cln3-Cdk1 kinase. (**A**) Western blot and quantification of the level of Igo1 phosphorylation (P-Igo1/total Igo1) during a synchronous cell cycle after elutriation of *IGO1-myc* cells (E4996). Mean (SD) of three experiments. Value was arbitrarily set at 1 for the first time point. (**B**) Igo1-S64 phosphorylation level in exponentially growing *cln1Δ cln2Δ cln3::GAL-CLN3* cells

*Figure 2 continued on next page*

*Figure 2 continued*

(E5261) shifted from SC-Gal to SC-D for the indicated time, filtered after 30 or 50 min and put back in SC-Gal for 10 and 20 min. Quantification of phospho-S64-Igo1 shows the mean (SD) of three experiments. Value was arbitrarily set at 1 for the first time point.

The following figure supplements are available for figure 2:

**Figure supplement 1.** FACS profiles and determination of median cell size at budding

**Figure supplement 2.** TORC1 activity increases during G1 and drops in mitosis.

activity. In contrast, neither the entry nor the progression of S phase appeared affected in *cdc28-as1 igo1,2Δ* cells (*Figure 3—figure supplement 2*), suggesting that the critical substrates of Rim15-

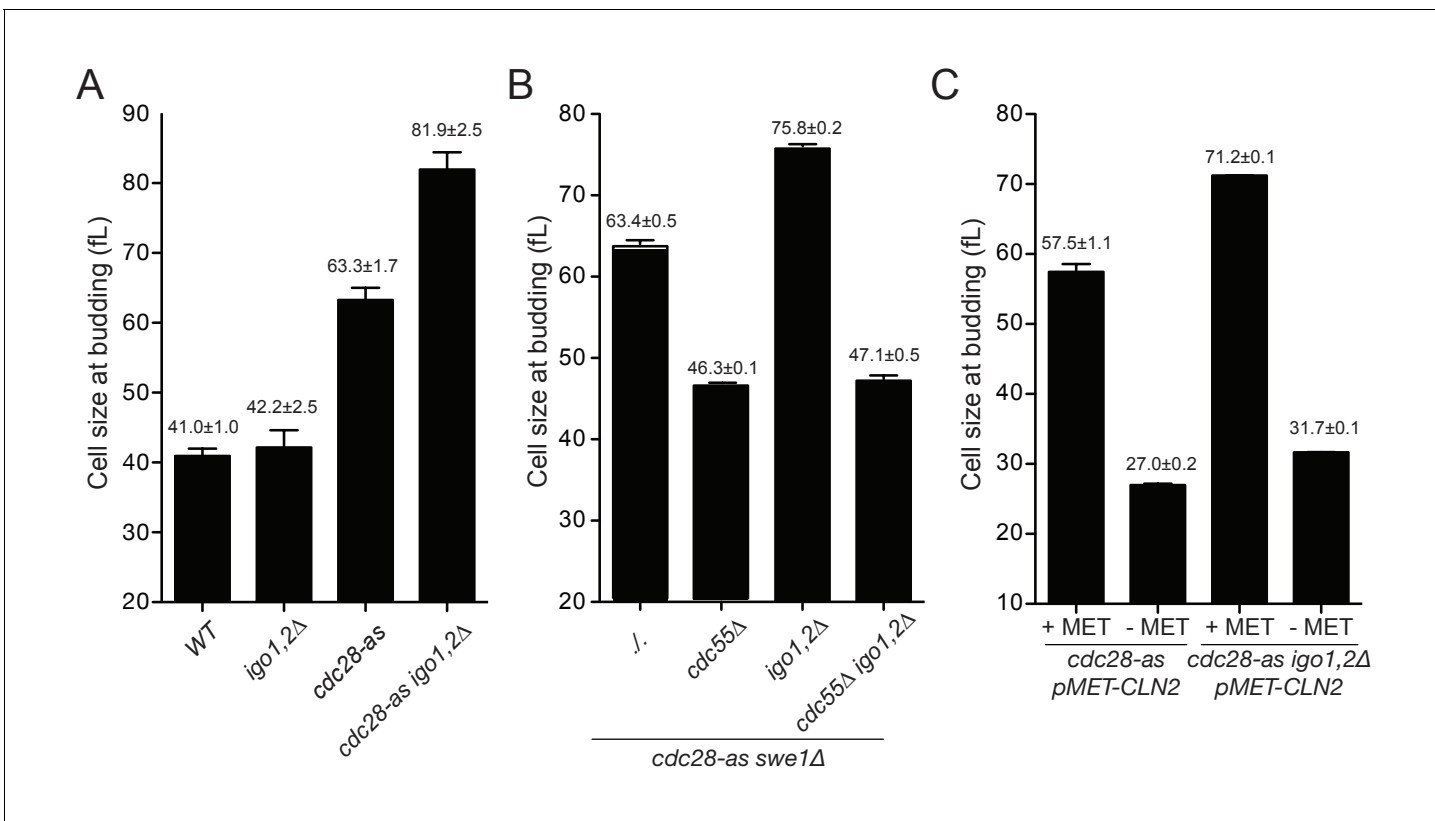

**Figure 3.** Rim15 and Cln-Cdk1 kinases cooperate to set the cell size at START. (**A**) Cells lacking Igo1,2 bud at a larger cell size. Elutriated early G1 cells of the indicated genotypes (E3087, E4259, E4479, E4458) were inoculated in SC-D medium at 30°C, and the population modal cell volume measured at the time of half-maximum budding. (**B**) The START delay of *cdc28-as1 igo1,2Δ* cells depends on *CDC55*. All strains (E4465, E5169, E4452, E4463) are also deleted for *SWE1*. (**C**) Ectopic *CLN2* expression from the *MET3* promoter suppresses the START delay of *cdc28-as1* and *cdc28-as1 igo1Δ igo2Δ* cells. Early G1 *cdc28-as1 pMET3-CLN2* (E5447) and *cdc28-as1 igo1Δ igo2Δ pMET3-CLN2* (E5159) cells were obtained by elutriation and put back in SC-D medium containing methionine (*CLN2* off) or not (*CLN2* on), and their size at budding measured as above. Cell volumes are indicated (fL) and are the mean (±SEM) of three independent elutriations for each strain.

The following figure supplements are available for figure 3:

**Figure supplement 1.** Rim15 and Cln-Cdk1 kinases cooperate to set the cell size at START.

**Figure supplement 2.** Cell cycle progression determined by FACS analysis of DNA content of elutriated early cells of the indicated genotype released in SC-D at 30°C.

Igo1,2-PP2A are those controlling START, not S-phase entry like for example Sic1 after rapamycin treatment (*Moreno-Torres et al., 2015*).

Next, we addressed whether the START delay of *cdc28-as1 igo1,2Δ* cells depends on PP2A$^{Cdc55}$. If this were the case, then *cdc28-as1 igo2Δ* cells containing Igo1-S64A that is unable to inhibit PP2A should have the same START delay as cells lacking Igo1,2 (*Bontron et al., 2013*; *Juanes et al., 2013*; *Sarkar et al., 2014*). Indeed *cdc28-as1 igo1-S64A igo2Δ* cells budded at 83.3 fL like *cdc28-as1 igo1,2Δ* (81.9 fL), compared to 63.3 fL for *cdc28-as1* cells (*Figure 3—figure supplement 1B*). Conversely, deleting *CDC55* that encodes the B-regulatory subunit of PP2A should suppress the large START size of *cdc28-as1 igo1,2Δ* cells. Since *CDC55* deletion leads to pleiotropic effects that can be suppressed by deleting *SWE1* (*Godfrey et al., 2017*; *Harvey et al., 2011*), we compared cell size at START after elutriation in *cdc28-as igo1,2Δ* cells also deleted for *SWE1*. We found that the START delay of *cdc28-as1 igo1,2Δ* cells (75.8 fL) was entirely dependent on *CDC55,* with *cdc28-as1 igo1,2Δ cdc55Δ* cells budding at 47.1 fL like *cdc28-as1 cdc55Δ* (46.3 fL) (*Figure 3B*). Thus in conditions of low CDK activity, the Rim15-Igo1,2 pathway promotes START by inhibiting PP2A$^{Cdc55}$.

To see if the increased START size of *cdc28-as1 igo1,2Δ* cells is due to a defect or delay in activating the transcriptional program responsible for the sudden burst of Cln1,2 cyclins at START, we tested whether this delay is suppressed by ectopic expression of Cln2. Expression of *CLN2* from the *MET3* promoter fully rescued (from 71.2 to 31.7 fL, *Figure 3C*) the START delay of *cdc28-as1 igo1,2Δ* cells and also suppressed the large size at budding of *cdc28-as1* cells growing without 1-NMPP1 (from 57.5 to 27.0 fL). Similar results were obtained with ectopic *CLN3* expression (*Figure 3—figure supplement 1C*). Altogether, our results suggest that the *cdc28-as1* allele is defective in size control due to insufficient or delayed transcriptional activation of *CLN1,2* genes and that the Rim15-Igo1,2 pathway inhibiting PP2A$^{Cdc55}$ somehow favors *CLN1,2* transcription.

## Timely Whi5 phosphorylation depends on *IGO1,2*

*CLN1,2* transcription depends on SBF, which is repressed by Whi5 and Stb1 in early G1. SBF inhibition is relieved towards the end of G1, when these repressors are progressively phosphorylated by Cln3-Cdk1 and displaced from SBF-dependent promoters (*Wang et al., 2009*). One explanation for the START delay of *cdc28-as1 igo1,2Δ* cells is that Whi5 might remain unphosphorylated for longer due to the combined effects of lowered Cln3-Cdk1 activity in *cdc28-as1* cells and lack of PP2A inhibition by Igo1,2. We tested this hypothesis by monitoring the level of Whi5 phosphorylation relative to cell size as elutriated early G1 *cdc28-as1* and *cdc28-as1 igo1,2Δ* cells progressed through G1 towards S phase. We found that Whi5 became hyper-phosphorylated when cells reached a median cell volume of 53 fL when Igo1,2 are present, but only at 74 fL when they are not (*Figure 4A*; *Figure 4—figure supplement 1*). Quantification of phospho-Whi5 confirmed that half-maximal phosphorylation was attained in cells ~ 20 fL smaller when Igo1,2 are present than in cells devoid of them (*Figure 4B*). Similar results were obtained when monitoring Stb1 phosphorylation (data not shown). We thus conclude that the timely phosphorylation of Whi5 and Stb1 in late G1 depends on the presence and activity of Rim15-Igo1,2, which prevent the dephosphorylation of Whi5 by inhibiting PP2A$^{Cdc55}$.

To confirm that Rim15 and Igo1,2 modulate START by controlling the timing of Whi5 phosphorylation, and not by another mechanism, we tested if the START delay of *cdc28-as1 igo1,2Δ* cells depends on the presence of Whi5 and its co-repressor Stb1. As before we elutriated and measured the size at half-maximal budding of *cdc28-as1 igo1,2Δ* cells deleted or not for *WHI5* and/or *STB1*. Deletion of *WHI5* alone lowered the size at budding of *cdc28-as1 igo1,2Δ* cells from 82 to 70 fL, whereas deletion of both *WHI5* and *STB1* reduced this size further to 63 fL, identical to that of *cdc28-as1* cells (*Figure 4C*). Deletion of *STB1* alone had little or no effect. We conclude that Whi5 and Stb1 are fully responsible for the START delay of *cdc28-as1 igo1,2Δ* cells, consistent with our proposal that the Rim15-Igo1,2 pathway controls START in conditions of low CDK activity by preventing excessive dephosphorylation of Whi5 and Stb1 by PP2A$^{Cdc55}$. It would be important, in future investigations, to test biochemically that these proteins are indeed dephosphorylated by PP2A$^{Cdc55}$.

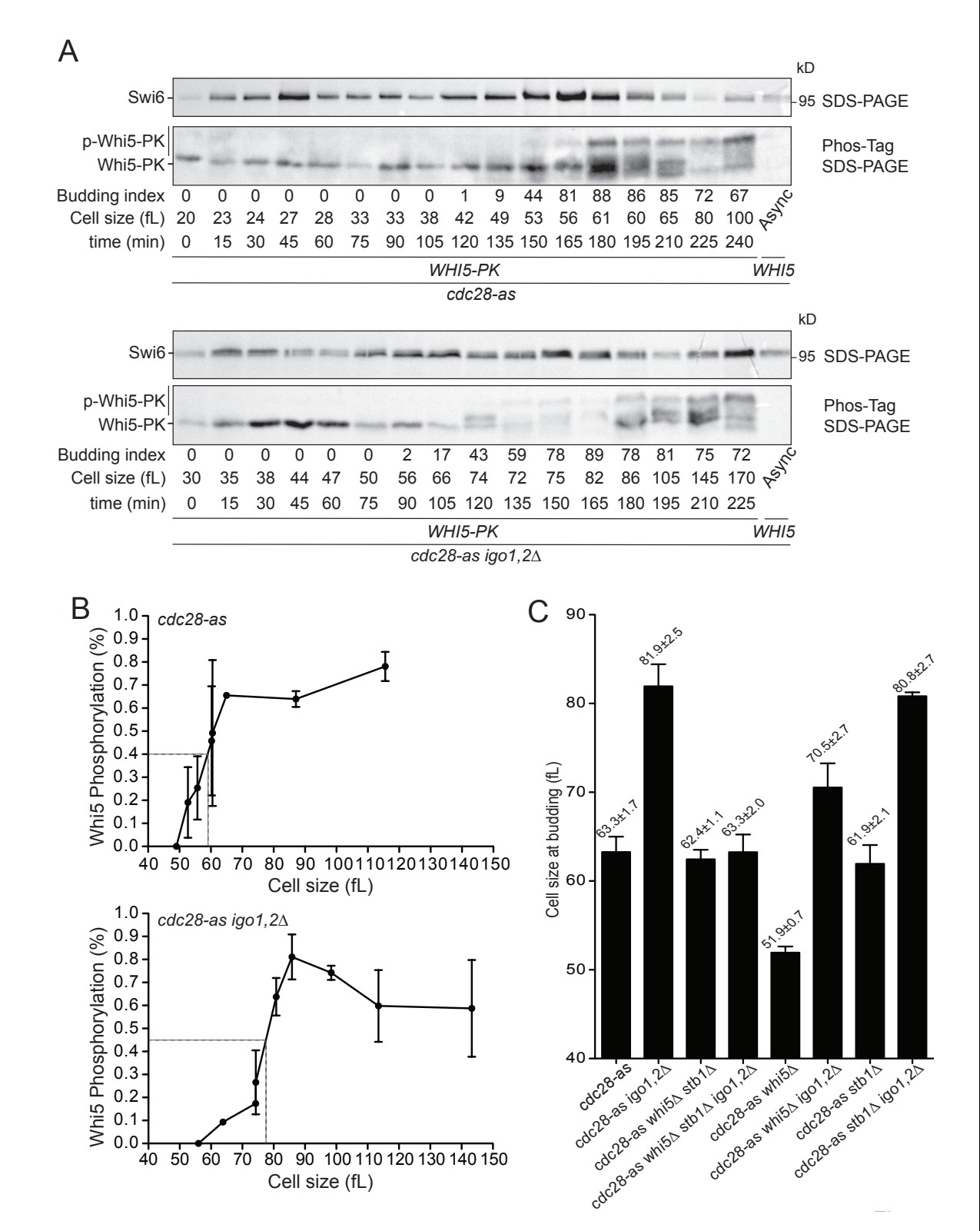

**Figure 4.** The Rim15 pathway controls START by promoting Whi5 phosphorylation. **(A)** Whi5 is phosphorylated at a larger cell size when Igo1,2 are missing. *cdc28-as1* (E5443) and *cdc28-as1 igo1,2Δ* (E5441) containing PK-tagged Whi5 were elutriated and Whi5 phosphorylation timing determined on Phos-tag gels. Untagged *WHI5* strain is shown as negative control for antibody specificity. **(B)** Quantification of Whi5 phosphorylation from Phos-tag western blots. Mean (SD), n = 3. **(C)** The START delay of *cdc28-as1 igo1,2Δ* cells is dependent on *WHI5* and *STB1*. Cells of the indicated genotype
*Figure 4 continued on next page*

*Figure 4 continued*

(E4479, E4458, E5222, E5218, E5188, E5227, E5189, E5221) were elutriated, early G1 cells incubated in SC-D at 30°C and the mean cell size at half maximum budding measured. Mean ±SD is indicated, n = 4.

The following figure supplements are available for figure 4:

**Figure supplement 1.** Cell size at budding after elutriation of *cdc28-as1 WHI5-3PK* (E5443) and *cdc28-as1 igo1,2Δ WHI5-3PK* (E5441) cells grown in SC-D.

**Figure supplement 2.** Cell cycle progression determined by FACS analysis of DNA content of elutriated early cells of the indicated genotype released in SC-D at 30°C.

## Rim15-Igo1,2 control START and contribute to chromosome stability after the diauxic shift

Until now, we described a role at START for the Rim15-Igo1,2 pathway only in conditions where Cdk1 activity was artificially lowered using the hypomorphic *cdc28-as1* allele. Interestingly, we found that this pathway is hyperactive in early G1 cells, when Cln3-Cdk1 activity is low. We therefore wondered whether Rim15-Igo1,2 might have a more prominent role in situations where Cln3 activity is physiologically low. *CLN3* translation decreases due to interference by a short upstream reading frame when nitrogen sources are limiting (*Gallego et al., 1997*; *Polymenis and Schmidt, 1997*). Moreover, it is known that *CLN3* mRNA, Cln3 protein and Cln3-Cdk1 kinase activity are drastically down-regulated when cells grow on poor carbon source, such after the diauxic shift when all fermentable carbohydrates in the medium have been utilized (*Hall et al., 1998*; *Parviz and Heideman, 1998*). We therefore tested whether the Rim15 pathway is activated and whether it controls START in wild type cells growing in YEP medium containing glycerol/lactate instead of glucose. *Figure 5A* shows that the fraction of phosphoS64-Igo1 increased from 20% to 64% when cells were shifted to poor carbon sources. These cells also passed START at a larger size (41 fL instead of 31 fL) when Rim15 signaling was abolished using a non-phosphorylable Igo1-S64A protein (*Figure 5B*). This demonstrates that the Rim15 pathway becomes hyperactive and controls START when cells grow by oxidative phosphorylation in poor carbon conditions.

To see if loss of the Rim15-Igo1,2 signaling pathway has consequences for genome stability, we measured the loss rate of an artificial ring chromosome in *WT*, *cdc28-as1* and *cdc28-as1 igo1,2Δ* cells grown in glucose or glycerol/lactate. Chromosome loss was increased 1.6-fold and 4.4-fold when the *IGO1,2* genes were deleted in *WT* or *cdc28-as1* cells growing in glucose, respectively. Strikingly chromosome loss increased 10.7-fold when cells were grown in the poorer carbon source (*Figure 5C*). This indicates that the Rim15 pathway becomes important for proper chromosome maintenance when cells grow at a lower rate, likely by inhibiting PP2A$^{Cdc55}$ to set an optimal size for START, although we cannot rule out that PP2A targets other than Whi5-Stb1 are involved. Interestingly Rim15 overexpression, which lowers the size of wild type, *cdc28-as1* and *cln3Δ* cells grown in rich carbon sources (*Figure 1—figure supplement 2*), also increases chromosome loss in *cdc28-as1* cells (*Figure 5D*). Thus, both ablation and hyper-activation of the Rim15 pathway controlling cell size at START impacts on chromosome maintenance in cells with low G1 CDK activity, suggesting that Rim15 is crucial for the proper balance between kinase and phosphatase activities at cell cycle entry.

## Discussion

### Rim15/Gwl disrupts the balance between kinases and phosphatases

Cell cycle progression is a tug of war between kinases and phosphatases. The fact that CDK activity is constantly opposed by phosphatases is witnessed by the rapid dephosphorylation of many target proteins *in vivo* when Cdk1 is chemically inhibited (*Holt et al., 2009*). Yet this balance between kinases and phosphatases has to be tipped over for cells to move from one metastable cell cycle state to the next. It has been well documented that the major cell cycle transitions are triggered by a sudden rise in CDK activity. In yeast, for example, cell cycle entry occurs when the Cln1,2-Cdk1 positive feedback loop leads to more *CLN1,2* transcription, S-phase entry when the Clb-Cdk1 inhibitor Sic1

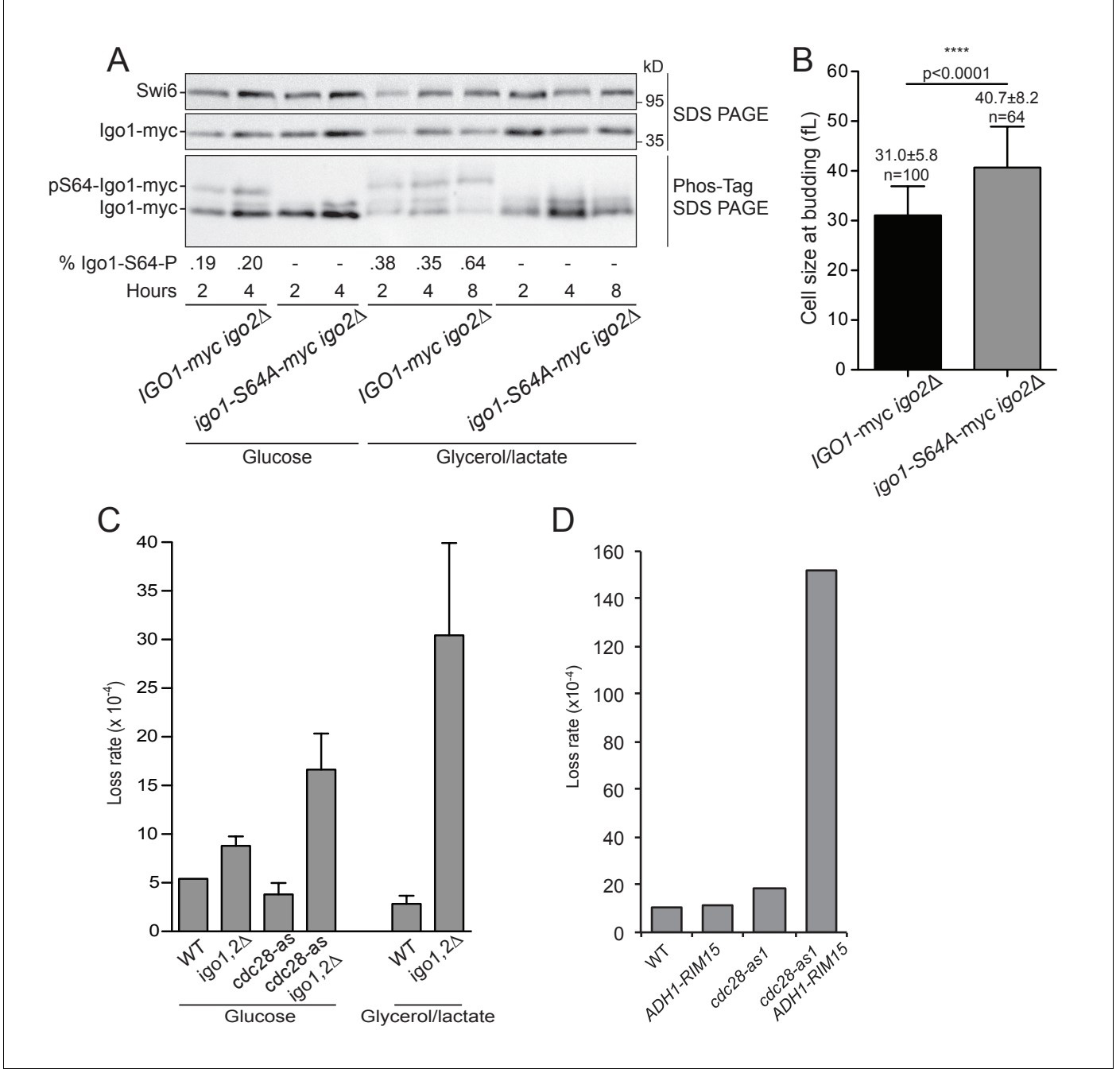

**Figure 5.** The Rim15 pathway is activated after the diauxic shift where it contributes to genome stability. (**A**) The level of Igo1 phosphorylation in *IGO1-myc* (E4974) and *igo1S64A-myc* (E4975) cells growing either on glucose or glycerol-lactate was determined on Phos-tag gels. Cells were shifted for 2 to 8 hr to YEPD (Glucose) or YEPGL (Glycerol/lactate) and whole cell protein extract (10 µg) analysed by SDS-PAGE or Phos-tag-PAGE. Cells grew longer in YEPGL (8 hr) so to reach the same cell density as in 4 hr in YEPD. The fraction of phospho-S64-Igo1, shown below the gel, was quantitated as in *Figure 2*. Swi6, loading control. (**B**) Cells containing non-phosphorylatable Igo1 pass START at a larger size when grown in respiratory conditions. *IGO1* and *igo1-S64A* cells grown in YEP-Glycerol/lactate were fixed, imaged and their size at budding determined using BudJ software. Mean cell volume ± SEM, n = 100 and n = 64, respectively; Student t test. (**C**) Chromosome loss in *igo1,2Δ* mutants. Cells of the indicated genotype (E4989, E4990, E5003, E5001) containing a 61 kb circular chromosome harbouring three ARSs (RCIII-3ARS) were grown for 6–8 generations without selection and then plated to determine the fraction of cells having lost RCIII (red colonies). Cells were grown either in SC-D (left) or in YEP-Lactate/Glycerol (right). Numbers indicate loss rate /cell • generation (Mean ± SEM, n = 3, 5–8000 colonies/sample). (**D**) Chromosome loss in *ADH1-RIM15* cells. Cells of shown genotype (E4989, E5497, E5003, E5498) were grown in SC-D for 12–15 generations and plated on sectoring medium to score red colonies having lost the RCIII chromosome, as in (**C**).

is suddenly destroyed, and G2/M when mitotic kinases stimulate their own activation. However, it is less clear how these feedback loops are initiated in the first place. Here, we identify the Rim15-Igo1,2 pathway, orthologous to metazoan Gwl-Arpp19/ENSA, as a new player in G1 progression in budding yeast. We show that Rim15-Igo1,2 transiently inhibits the phosphatase PP2A$^{Cdc55}$ in early G1 cells, helping them to tip the balance towards more pronounced phosphorylation of Whi5 by Cln3-Cdk1. Deletion of *IGO1* and *IGO2* has subtle phenotypes in wild type cells, such as a slightly larger size at START and heightened chromosome loss, but these phenotypes are exacerbated when G1 CDK activity is low, such as in *cdc28-as1* cells or wild type cells growing on poor carbon sources. Thus, decreased Cln3-Cdk1 activity combined to lack of PP2A$^{Cdc55}$ inhibition in *igo1,2Δ* cells leads to delayed Whi5 phosphorylation and budding at a larger cell size. This larger size at budding is suppressed by ectopic *CLN2* or *CLN3* expression and by deletion of *WHI5* and *STB1*, which are known targets of Cln3-Cdk1 (*Costanzo et al., 2004*; *de Bruin et al., 2004*; *Holt et al., 2009*). This demonstrates that the Rim15-Igo1,2 pathway promotes START by activating SBF-dependent transcription. That Rim15 does so by inhibiting PP2A$^{Cdc55}$ is supported by finding that the large size at budding of *cdc28-as1 igo1,2Δ* cells is suppressed by deletion of *CDC55*, but not by an Igo1-S64A version that cannot inhibit PP2A. The inhibition of PP2A by Endosulfines/Igo1,2 proteins was demonstrated previously *Xenopus* egg extracts (*Gharbi-Ayachi et al., 2010*; *Mochida et al., 2010*), but also in yeast cells stressed by heat (*Juanes et al., 2013*), rapamycin (*Bontron et al., 2013*) or nutrient deprivation at meiotic entry (*Sarkar et al., 2014*). We provide here the first evidence that Rim15 controls START during vegetative growth. It is tempting to speculate that analogous mechanisms might control G1 progression in mammals where Rb family members, like Whi5 in yeast, are dephosphorylated by PP2A-B55 (*Kurimchak and Graña, 2015*).

The Rim15 analogs Gwl, Mastl and Ppk18 have been shown to regulate entry, progression and exit of mitosis in flies, frog egg extracts, mammals and fission yeast (*Álvarez-Fernández et al., 2013*; *Chica et al., 2016*; *Cundell et al., 2013*; *Gharbi-Ayachi et al., 2010*; *Mochida et al., 2010*; *Yu et al., 2004*). In the absence of Gwl or ENSA/Arpp19, cells fail to reach or maintain the level of CDK phosphorylation required for an efficient mitosis. Recently, it was shown that PP2A-B55 and CycB-Cdk1 can promote M-phase entry by forming two bi-stable switches that are interlocked via their antagonistic effects on Gwl-ENSA (*Mochida et al., 2016*). A central tenet of this model is that Gwl is activated by CycB-Cdk1 to inhibit its counteracting phosphatase. However Gwl might also be active also outside of mitosis, as recently suggested by the severe S-phase progression defects following ENSA depletion in human cells, due to decreased half-life of Treslin/TICRR (*Charrasse et al., 2017*). Gwl also plays a role in *S. pombe* sexual differentiation coupled to nutrient availability, by interfacing TORC1 and TORC2 activities (*Martín et al., 2017*). Altogether, it appears that transient inhibition of PP2A by the Gwl pathway can be used in various cell cycle stages and physiological situations.

## Rim15, a nutrient-controlled starter for START in budding yeast

In contrast to metazoans, inactivation of the Rim15/Gwl pathway has no obvious cell cycle phenotype in budding and fission yeasts grown in rich medium. This is because both Rim15 and Ppk18 are repressed by TORC1/PKA when nutrients are plentiful and proliferation rates high (*Chica et al., 2016*; *Pedruzzi et al., 2003*). When nutrients are scarce or when TORC1 is inhibited by rapamycin, Rim15/Ppk18 is activated, Igo1,2 are phosphorylated and PP2A$^{Cdc55/Pab1}$ activity is down-regulated to favor cell differentiation programs such as gametogenesis or quiescence (*Bontron et al., 2013*; *Martín et al., 2017*; *Sarkar et al., 2014*; *Talarek et al., 2010*). Rim15/Ppk18 also has an effect on the cell cycle by promoting precocious mitotic entry in fission yeast to adapt cell size to nitrogen availability (*Chica et al., 2016*), or to promote G$_0$ arrest by stabilizing the CDK inhibitor Sic1 (*Moreno-Torres et al., 2015*). Here, we show that Rim15-Igo1,2 controls cell cycle entry (START) in budding yeast when cells are grown in conditions of low CDK activity. Accordingly, Rim15 activity measured by Igo1 phosphorylation increases upon Cdk1 inhibition and decreases when *CLN3* is overexpressed. Thus, Rim15 is negatively regulated by G1-CDK in yeast, consistent with the decreased Rim15 phosphorylation and increased Igo1,2 phosphorylation observed in *cdc28-as1* cells briefly exposed to 1-NMPP1 (*Holt et al., 2009*). At first glance, this finding seems at odds with the activation of Gwl by CDK at mitotic entry in metazoans, but can be explained by the multiple and partly overlapping regulation of Gwl in different organisms. The residues important for Gwl activation by M-CDK in *Xenopus* (T193, T206) and mammals and the auto-activation site (S883) are all

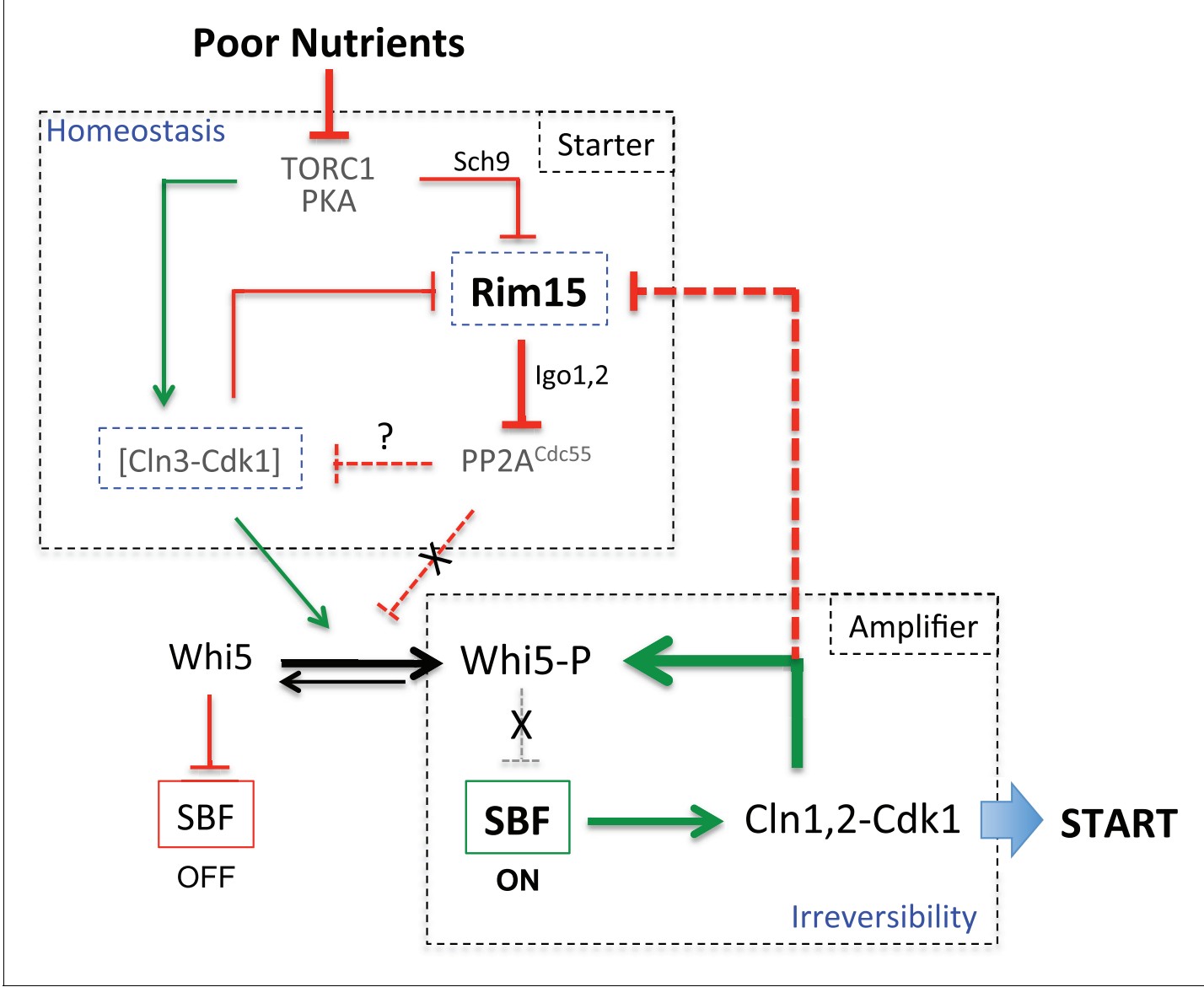

**Figure 6.** Model for homeostatic control of START by antagonism between Cln3-Cdk1 and Rim15, and cooperation for Whi5 phosphorylation. TORC1, PKA and PP2A$^{Cdc55}$ have additional targets while Cln3-Cdk1 also targets Stb1, which are not shown here for simplicity. When nutrients are poor, TORC1 is down regulated leading to low Cln3 levels and higher Rim15 activity. High Rim15 then inhibits PP2A$^{Cdc55}$ to promote Whi5 phosphorylation and START despite low Cln3-Cdk1 activity. Conversely, when nutrients are rich, high TORC1 leads to Cln3 accumulation, Rim15 inhibition and high PP2A$^{Cdc55}$ activity. The regulatory network leading to START and cell cycle entry can be decomposed in two modules: the Starter module (on top) favours initial Whi5 phosphorylation by decreasing PP2A activity when Cln3-Cdk1 activity is low, is coupled to nutrient availability and provides cell size homeostasis; the Amplifier module (on bottom) brings in positive feedback by Cln1,2-Cdk1 for full Whi5 phosphorylation and irreversibility of the START transition. Cln3-Cdk1 is inactivated by dephosphorylation (*Tyers et al., 1992*), possibly by PP2A. Negative feedback between Cln3-Cdk1 and Rim15 in the homeostatic module maintains a proper balance between kinase and phosphatase activities acting on Whi5.

missing in yeast Rim15 (*Blake-Hodek et al., 2012*). Conversely, Rim15 contains sites (T1075, T1565) that are phosphorylated by CDKs to regulate its nuclear exclusion and cytoplasmic retention. It has been shown that Rim15 and Igo1,2 concentrate in the nucleus after phosphate starvation, after the diauxic shift or when TORC1 is inactivated, as cells accumulate in G1 (*Mirisola et al., 2014*). Upon phosphate re-addition Pho80-85 (a CDK-like kinase) phosphorylates Rim15 to take it out of the nucleus (*Wanke et al., 2005*). The same site phosphorylated by Pho80-85 is also phosphorylated by Cln2-Cdk1 (*Breitkreutz et al., 2010*; *Holt et al., 2009; Moreno-Torres et al., 2017*). We, therefore,

propose that decreasing CDK activity in budding yeast causes Rim15 to accumulate in the nucleus, where it can phosphorylate Igo1,2 proteins and inhibit PP2A$^{Cdc55}$ against its nuclear targets.

Interestingly, we found that Rim15 activity and Igo1 phosphorylation increase markedly when yeast cells undergo the diauxic shift and rely on mitochondrial oxidative phosphorylation. This transition is accompanied by a stark reduction in Cln3 levels and associated kinase activity (*Hall et al., 1998*; *Parviz and Heideman, 1998*). These cells also adjust their cell cycle to the poorer carbon source by lowering the Cln3 threshold required for START, which reduces their overall cell size (*Ferrezuelo et al., 2012*). How cells growing in poor nutrients reduce their critical cell size for START despite reduced protein synthesis rates and low Cln3 levels has been a long standing mystery (*Jorgensen and Tyers, 2004*). We show here that Rim15 and Igo1,2 are required for the reduced START size of cells growing on non-fermentable carbon sources, by protecting Whi5 from being excessively dephosphorylated by PP2A$^{Cdc55}$.

## Size homeostasis by negative feedback of Cln3-Cdk1 on Rim15

Homeostatic systems require a sensor that negatively feeds back to an effector mechanism controlling the output of a system, so to keep the variable being measured within an acceptable range. The early G1 cyclin Cln3 has attributes for a sensor that controls the critical size at START in yeast: its abundance is coupled to growth rate; it binds Cdk1, phosphorylates Whi5 and releases the inhibition of the SBF transcription factor responsible for cell cycle commitment. Here, we demonstrate that the Rim15 kinase also promotes SBF activation, by inhibiting the protein phosphatase (PP2A$^{Cdc55}$) that counteracts Cln3-Cdk1's action on Whi5. But importantly, it does so only in conditions of low Cln3-Cdk1 activity. Indeed deletion of *RIM15* or *IGO1,2* has little or no effect on size when cells contain normal levels of Cln3 (WT cells grown in rich medium), but increases the size at budding in cells containing lower Cln3-Cdk1 activity (*cdc28-as1* cells grown in rich medium, or WT cells grown on poor carbon sources). Furthermore, we show that the converse is also true: cells in which Cln3 expression is induced respond by rapidly shutting off Rim15 kinase activity on Igo1,2. Hence, Cln3-Cdk1, Rim15 and PP2A$^{Cdc55}$ form a homeostatic system correcting the size at START when Cln3 levels fluctuate, through negative feedback of Cln3-Cdk1 on Rim15: when Cln3 is high, Rim15 activity is low and PP2A$^{Cdc55}$ counteracts the effect of high Cln3; when Cln3 is low, Rim15 activity increases, PP2A is inhibited, thus lowering the Cln3 threshold for multisite Whi5 phosphorylation and START (*Figure 6*). Besides negative regulation by Cln3-Cdk1, Rim15 is also repressed by the TORC1 and PKA pathways, which control cell proliferation in response to nutrient availability. TORC1 and PKA signaling decreases when nutrients are limiting, which activates Rim15 and facilitates START. The current 'Cln3 only' model of START control was unable to explain why cells growing on poor nutrients and containing little Cln3 do in fact pass START at a smaller, not larger cell size (*Jorgensen and Tyers, 2004*; *Turner et al., 2012*). One advantage for starved cells to be smaller might be that small cells have a greater surface area to volume ratio, which optimizes the yield of cell surface nutrient transporters relative to cell volume. We propose that the Rim15-Igo1,2 pathway is the missing component that facilitates START in poor nutrient conditions, compensating for low Cln3 activity in starved cells and making them pass START at a smaller size.

Several models have been proposed to explain why yeast cells pass START at a precise size despite a constant nuclear Cln3 concentration (*Schmoller and Skotheim, 2015*). However, none considered that Whi5 has to be phosphorylated on many sites to trigger START and that inhibition of the counteracting phosphatase could tip the balance towards full Whi5 phosphorylation. Our results suggest this is the case. We found that Rim15 activity is highest in early G1 cells, even when wild type cells are grown in rich medium, at a time when Cln3 levels are minimal (*Thorburn et al., 2013*; *Zapata et al., 2014*). Cln3 levels and Rim15 activities are under control of TORC1 and nutrient availability. TORC1 activity responds rapidly to changes in available amino acids (*Loewith and Hall, 2011*). We tried to measure TORC1 activity along G1 using Rps6 (ribosomal S6 protein) phosphorylation as a proxy (*González et al., 2015*; *Yerlikaya et al., 2016*) and found evidence that it might increase during G1 (*Figure 2—figure supplement 2*), consistent with the decrease in Rim15 activity during G1. In early G1 cells Rim15 would promote the initial phosphorylation of Whi5 when Cln3 levels are low, and once SBF is activated Cln1,2 cyclins phosphorylate Whi5 further via positive feedback, triggering irreversible cell cycle commitment but also shutting off Rim15 activity (*Figure 6*).

Our discovery that the Rim15/Gwl pathway regulates Whi5 phosphorylation in yeast in response to nutrient quality and Cln3 levels leads to a more complete model for cell size homeostasis at

START. This pathway becomes particularly important when cells grow on poor nutrients or when they have exhausted glucose in the medium and rely on oxidative phosphorylation, as indicated by the START delay and 10-fold increased chromosome loss when *igo1,2Δ* cells are grown in glycerol/lactate. Conversely, Rim15 overexpression drives cells into entering the cell cycle even when nutrients are limiting, which might lead to defects in chromosome replication and/or segregation. Future studies will tell whether this scenario holds in mammalian cells where Gwl and CycD are frequently overexpression in cancer cells (*Otto and Sicinski, 2017*; *Vera et al., 2015*). That Rb, p107 and p130, the mammalian orthologs of Whi5, are dephosphorylated by PP2A-B55 in interphase opens the possibility that Gwl might also control G1 progression in human cells (*Kurimchak and Graña, 2015*).

## Materials and methods

### Yeast strains, plasmids and culture

All strains used are listed in *Table 1*. They were obtained by standard genetic methods and are all congenic or backcrossed at least four times to W303 (*MATa ade2-1 trp1-1 can1-100 leu2-3,112 his3-11,15 ura3-1*). The *IGO1-myc8* and *igo1-S64A-myc8* strains were made by integrating EcoRV–linearized plasmids D2107 and D2106 at the *TRP1* locus, respectively. D2107 and D2106 were built by cloning a HindIII-SacI fragment from pLC1427 and pLC1430, respectively, into YIplac204 vector (*Gietz and Sugino, 1988*). The fragments obtained from pLC1427 or pLC1430 contain 490 nt of the promoter and the wild type *IGO1* or *igo1-S64A* ORF, respectively, and 8 in-frame myc tags. The Whi5-sfGFP and the Whi5-PK3 strains were constructed using published protocols (*Khmelinskii et al., 2011*; *Knop et al., 1999*). Cells were grown in supplemented minimal medium (SC) or in rich medium (YEP) as indicated, to which glucose (D), or raffinose (R), or galactose (G), or lactate, or glycerol was added to 2% final unless stated otherwise. For chromosome loss assay, the sectoring medium was prepared with 1% Yeast Extract, 2% Bactopeptone, 2% Agar (Difco, BD, Franklin Lakes, NJ), 4% Glucose.

### Centrifugal elutriation

For centrifugal elutriation, cells were inoculated at $1 \times 10^5$ cells/mL and grown overnight in 1–2 liters of YEPR at 30°C until $5.10^7$ cells/mL. Cells were harvested by centrifugation and resuspended in 400 ml of cold YEPRaff, and sonicated 30 s on VibraCell 72405 sonicator set at amplitude 60. Cells were kept on ice, loaded onto a Beckman J6-MC centrifuge (JE5.0 rotor) at 3600 rpm, with the pump (Masterflex, Cole Parmer, Barrington, IL) flow rate set at 20 mL/min until the 40 mL chamber was full, and left to equilibrate for 10 min with cold SC-D medium. Pump speed was then gradually increased by 0.02 increments until small daughter cells were elutriated (between 22 and 25 mL/min). Cells were collected until a sufficient amount was reached. For *MET3p-CLN2* cells were elutriated in SC-D in which the methionine was omitted and then released in SC-D with or without methionine. For *GAL10p-CLN3* cells were elutriated in SC-R and then released in SC-D, SC-R + G00.1% and SC-G. Modal cell volume was measured with a Cell Analyser System (CASY1 TTC, Schärfe System, Reutlingen, Germany). Cell cycle progression was determined by measuring budding index and DNA content using Sytox Green and FacsCalibur cytometer (Becton Dickinson, France).

### Cell cycle phase duration and median cell volume

Cells were inoculated at $5.10^5$ cells/mL in SC-D. Median cell volume and cell concentration were measured every hour for 10 hr using a cell counter (CASY1 TTC). To discard debris and cell clumps the median cell volume was determined from cells spanning 2.5 to 10 μm in diameter. Population doubling time was calculated by linear regression during exponential growth. The fraction of cells in G1, S and G2+M was determined as follows: every hour 1 mL of cells was pulsed for 10 min with 25 μM EdU. EdU incorporation was quantified using a Click reaction and FACS as described (*Talarek et al., 2015*). The fraction of cells in G1, S and G2+M phases was determined from PI-EdU dot blots. The duration of each phase was then calculated by multiplying the doubling time by the fraction of cells in each cell cycle phase.

**Table 1.** Yeast strains.

| Name | Genotype |
| --- | --- |
| **E3087** | ***MATa, URA3::GPD-TK(5x), AUR1c::ADH-hENT1*** |
| E4259 | *MATa, igo1::NatNT, igo2::KanMX, URA3::GPD-TK(5x), AUR1c::ADH-hENT1* |
| E4331 | *MATa, cdc28-as1(F88G), igo2::KanMX, URA3::GPD-TK(5x), AUR1c::ADH-hENT1* |
| E4452 | *MATa, cdc28-as1(F88G), swe1::LEU2, URA3::GPD-TK(5x), AUR1c::ADH-hENT1* |
| E4458 | *MATa, cdc28-as1(F88G), igo1::NatNT, igo2::KanMX, URA3::GPD-TK(5x), AUR1c::ADH-hENT1* |
| E4463 | *MATa, cdc28-as1(F88G), swe1::LEU2, igo1::NatNT, igo2::KanMX, URA3::GPD-TK(5x), AUR1c::ADH-hENT1* |
| E4465 | *MATa, cdc28-as1(F88G), swe1::LEU2, cdc55::TRP1, URA3::GPD-TK(5x), AUR1c::ADH-hENT1* |
| E4471 | *MATa, cdc28-as1(F88G), rim15::KanMX, URA3::GPD-TK(5x), AUR1c::ADH-hENT1* |
| E4479 | *MATa, cdc28-as1(F88G), URA3::GPD-TK(5x), AUR1c::ADH-hENT1* |
| E4948 | *MATα, igo1::NatNT, igo2::KanMX, WHI5-sfGFP* |
| E4974 | *MATa, igo1::NatNT, igo2::KanMX, TRP1::IGO1-myc8, URA3::GPD-TK(5x), AUR1c::ADH-hENT1* |
| E4975 | *MATa, igo1::NatNT, igo2::KanMX, TRP1::IGO1-S64A-myc8, URA3::GPD-TK(5x), AUR1c::ADH-hENT1* |
| E4989 | *MATα, RCIII-SUP11-LEU2-3ARS* |
| E4990 | *MATα, igo1::NatNT, igo2::KanMX, RCIII-SUP11-LEU2-3ARS* |
| E4995 | *MATa, cdc28-as1(F88G), igo1::NatNT, igo2::KanMX, TRP1::igo1-S64A-myc8, URA3::GPD-TK(5x), AUR1c::ADH-hENT1* |
| E4996 | *MATa, cdc28-as1(F88G), igo1::NatNT, igo2::KanMX, TRP1::IGO1-myc8, URA3::GPD-TK(5x), AUR1c::ADH-hENT1* |
| E5001 | *MATα, cdc28-as1(F88G), igo1::NatNT, igo2::KanMX, RCIII-SUP11-LEU2-3ARS* |
| E5003 | *MATα, cdc28-as1(F88G), RCIII-SUP11-LEU2-3ARS* |
| E5100 | *MATα, WHI5-sfGFP* |
| E5101 | *MATα, cdc28-as1(F88G), WHI5-sfGFP, URA3::GPD-TK(5x), AUR1c::ADH-hENT1* |
| E5102 | *MATα, cdc28-as1(F88G), igo1::NatNT, igo2::KanMX, WHI5-sfGFP, URA3::GPD-TK(5x), AUR1c::ADH-hENT1* |
| E5159 | *MATa, cdc28-as1(F88G), igo1::NatNT, igo2::KanMX, TRP1::MET3-CLN2(1x)* |
| E5169 | *MATa, cdc28-as1(F88G), swe1::LEU2, cdc55::TRP1, igo1::NatNT, igo2::KanMX,* |
| E5188 | *MATα, cdc28-as1(F88G), whi5::KanMX, AUR1c::ADH-hENT1, URA3::mCherry-TUB1* |
| E5189 | *MATα, cdc28-as1(F88G), stb1::KanMX, URA3::GPD-TK(5x)* |
| E5218 | *MATa, cdc28-as1(F88G), whi5::KanMX, stb1::KanMX, igo1::NatNT, igo2::KanMX, URA3::GPD-TK(5x)* |
| E5221 | *MATα, cdc28-as1(F88G), stb1::KanMX, igo1::NatNT, igo2::KanMX, URA3::GPD-TK(5x)* |
| E5222 | *MATα, cdc28-as1(F88G), whi5::KanMX, stb1::KanMX, URA3::GPD-TK(5x)* |
| E5227 | *MATa, cdc28-as1(F88G), whi5::KanMX, igo1::NatNT, igo2::KanMX, URA3::GPD-TK(5x)* |
| E5261 | *MATa, igo1::NatNT, igo2::KanMX, TRP1::IGO1-myc8, cln1::hisG, cln2Δ, cln3::GAL10-CLN3::URA3* |
| E5231 | *MATa, cdc28-as1(F88G), igo1::NatNT, igo2::KanMX, cln3::GAL10-CLN3::URA3* |
| E5323 | *MATα, cdc28-as1(F88G), rim15::NatNT::ADH-yeGFP-RIM15* |
| E5373 | *MATa, cdc28-as1(F88G), cln3::GAL10-CLN3::URA3* |
| E5441 | *MATa, cdc28-as1(F88G), igo1::NatNT, igo2::KanMX, WHI5-3PK::HIS3Kl, URA3::GPD-TK(5x), AUR1c::ADH-hENT1* |
| E5443 | *MATa, cdc28-as1(F88G), WHI5-3PK::HIS3Kl, URA3::GPD-TK(5x), AUR1c::ADH-hENT1* |
| E5447 | *MATa, cdc28-as1(F88G), TRP1::MET3-CLN2(1x)* |
| E5492 | *MATa, cdc28-as1(F88G), cln3::GAL10-CLN3::URA3, URA3::GPD-TK(5x), AUR1c::ADH-hENT1* |
| E5493 | *MATa, cln3::GAL10-CLN3::URA3, URA3::GPD-TK(5x), AUR1c::ADH-hENT1* |

*Table 1 continued on next page*

*Table 1 continued*

| Name | Genotype |
|------|----------|
| **E3087** | **MATa, URA3::GPD-TK(5x), AUR1c::ADH-hENT1** |
| E5497 | MATa, rim15::NatNT::ADH-yeGFP-RIM15, RCIII-SUP11-LEU2-3ARS |
| E5498 | MATa, cdc28-as1(F88G), rim15::NatNT::ADH-yeGFP-RIM15, RCIII-SUP11-LEU2-3ARS |
| E5504 | MATa, rim15::NatNT::ADH-yeGFP-RIM15, URA3::GPD-TK(5x), AUR1c::ADH-hENT1 |
| E5505 | MATa, rim15::NatNT::ADH-yeGFP-RIM15, URA3::GPD-TK(5x), AUR1c::ADH-hENT1, cln3:: GAL10-CLN3::URA3 |

## Microscopy and BudJ analysis

For Whi5-sfGFP detection and cell size measurements (*Figure 3—figure supplement 1A*), cells exponentially growing in SC-D at 30°C were briefly sonicated and immediately imaged using a 63× HCX Plan Apochromat, 1.4 NA objective, with 8–10 z stacks of 0.5 µm, GFP filter cube, 200 ms acquisition time, and no binning on a Leica DM6000 microscope equipped with a CoolSNAP HQ2 camera (Roper Scientific, Germany) and MetaMorph 7.6 (Molecular Devices, Sunnyvale, CA) to obtain fluorescence and bright-field images. Z sections were projected to two-dimensional images and processed with ImageJ (W Rasband, NIH, https://imagej.nih.gov/ij/). Cell size of unbudded cells displaying Whi5-GFP in their nucleus was determined using BudJ (*Ferrezuelo et al., 2012*), with the following parameters: BF channel, Scale factor 0.1 micron/pixel, Max cell radius 10 microns, Max window for cell edge 1 micron, Min gray change at cell edge 30%. FL1 foci parameters, Min FL1 change at foci 10% Min foci size 2 pixels, Max foci size 500 pixels. For cell size measurements only (*Figure 5B*), cells were exponentially grown in (YEP-2% Glycerol/2% lactate) at 30°C, fixed with PFA 2% for 20 min, and imaged as described above. Cell size of small budded cells was determined using BudJ and the following parameters: BF channel, Scale factor 0.1 micron/pixel, Max cell radius 6 microns, Max window for cell edge 1 micron, Min gray change at cell edge 15%.

## Whole-cell protein extract from yeast cells

5–40 ml of yeast culture was centrifuged and resuspended in 1 mL cold TCA 20%. Cells were concentrated to 200 µL in 10% TCA, and 200 µL glass beads (Zirconia/Silica Beads; BioSpec Products, Inc., Bartlesville, OK) was added. Cells were then lysed using a Bullet Blender Storm 24 (Next Advance, Inc., Averill Park, NY). Supernatants were transferred to new tubes, beads washed twice with 200 µL TCA 10%, and extracts were pooled. Extracts were centrifuged at 3000 rpm for 10 min, and the pellet was resuspended in 30–200 µL Laemmli buffer containing 5–10 µl Tris base (1 M). The extracts were boiled 5 min, centrifuged 5 min at 12,000 rpm and transferred to a new tube. Protein concentration was determined by Bradford assay.

## SDS-PAGE and western blots

For Igo1-myc8, 10 µg whole-cell extracts were run on 12% acryl–bisacrylamide (29:1) gels and 8.5% acryl–bisacrylamide (29:1) 25 µM Phos-tag gels (Wako Chemicals GmbH, Germany). For Whi5-PK3, 10 µg whole-cell extracts were run on 12.5% acryl–bisacrylamide (125:1) gels and 8.5% acryl–bisacrylamide (29:1) 5 µM Phos-tag gels. Proteins were transferred to nitrocellulose membranes (Protran; Amersham, GE Healthcare, Germany) using wet blotting and revealed using standard immuno-blotting and ECL procedures. Western blot quantifications were performed from images acquired with the multi-application gel imaging system PXi 4 (Syngene, A Division of Synoptics Ltd, UK), using ImageJ. Primary antibodies were mouse monoclonal anti-myc (1:1000 from ascites for Igo1-myc8), rabbit polyclonal anti-swi6 (1:100,000) and mouse monoclonal anti-PK (1:5,000; clone SV5-Pk1; Bio-Rad Antibodies, France).

## Chromosome loss assay

A 3ARS ring chromosome III (RCIII) harboring *LEU2* and *SUP11-1* markers was used (*Dershowitz and Newlon, 1993*). Hence *ade2-1* cells containing RCIII form white colonies while

those having lost RCIII form red colonies (*Hieter et al., 1985*). Cells were first grown in selective minimum medium (SD without leucine), and a known number (X) of cells was plated on sectoring medium to determine the fraction of cells without RCIII in the starting culture (A= red colony number formed). Cells were then grown in non-selective medium (SC-D or YEPLG) for 6–15 generations (N), after which a defined number of cells (Y) were plated on sectoring medium. Red colonies were scored (B). The liquid culture loss rate was determined as follows ((B/Y)-(A/X))/N and is expressed as loss per cell per generation.

## Acknowledgements

We are grateful to M Aldea for help with BudJ analysis, to B Leggio for help with statistical analyses, to A Gonzalez for help with Rps6 phosphorylation assay, to S Piatti for critical reading of the manuscript, to C De Virgilio and M Tyers for sharing data prior to publication, and to members of the Castro/Lorca, Piatti and Schwob labs for many fruitful discussions. We thank MRI imaging facility for help with microscopy and FACS analysis. NT was supported by fellowships from the Fondation ARC pour la Recherche contre le Cancer (ARC) and European Union FP7 (Marie Curie IEF). This work was supported by grants from Agence Nationale de la Recherche (ANR) and Association pour la Recherche contre le Cancer (ARC) to ES.

## Additional information

### Funding

| Funder | Grant reference number | Author |
| --- | --- | --- |
| Fondation ARC pour la Recherche sur le Cancer | PJA 20151203349 | Etienne Schwob |
| Agence Nationale de la Recherche | ANR-14-CE11-0007-03 | Etienne Schwob |
| European Commission | 628961 ESCA-Y | Nicolas Talarek |
| Fondation ARC pour la Recherche sur le Cancer | PDF20121206166 | Nicolas Talarek |

The funders had no role in study design, data collection and interpretation, or the decision to submit the work for publication.

### Author contributions

NT, Data curation, Formal analysis, Supervision, Validation, Investigation, Visualization, Methodology, Writing—review and editing; EG, Data curation, Formal analysis, Investigation, Methodology, Project administration; ES, Conceptualization, Data curation, Supervision, Funding acquisition, Writing—original draft

### Author ORCIDs

Etienne Schwob, http://orcid.org/0000-0002-9369-6419

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
