## [Decision Letter]

Thank you for submitting your article "Homeostatic control of START through negative feedback between Cln3-Cdk1 and Rim15/Greatwall kinase in budding yeast" for consideration by *eLife*. Your article has been favorably evaluated by Randy Schekman (Senior Editor) and three reviewers, one of whom, Andrea Musacchio (Reviewer #1), is a member of our Board of Reviewing Editors. The following individuals involved in review of your submission have agreed to reveal their identity: Michael L Goldberg (Reviewer #2); Frank Uhlmann (Reviewer #3).

The reviewers have discussed the reviews with one another and the Reviewing Editor has drafted this decision to help you prepare a revised submission.

Summary:

In this manuscript, Talarek et al. investigate the contribution of the Greatwall-Endosulfine pathway to budding yeast cell size regulation. Greatwall-Endosulfines are inhibitors of the major cell cycle phosphatase PP2A-Cdc55 and they have so far mainly been studied as positive regulators of mitotic entry in metazoan cells. Being major phosphatase regulators, it is conceivable that their role extends beyond mitotic regulation, but this possibility has not been widely explored. This study demonstrates a role for the budding yeast Greatwall-Endosulfines, Rim15-Igo1/2, in regulating the duration of G1 and thereby cell size. The authors find that in rich growth medium, Rim15-Igo1/2 are only transiently active in small G1 daughter cells and have limited impact on cell cycle progression. In contrast, if nutrients are restricted, Rim15-Igo1/2 are activated to inhibit PP2A and thereby facilitate cell cycle entry at small cell size. This provides a molecular explanation for how nutrient signalling is linked to cell size control in budding yeast. Given the conservation of these components, this regulation likely applies to other organisms. This fundamentally extends our understanding of both the Greatwall-Endosulfine pathway and of cell size control. All three reviewers found the manuscript interesting and recommended publication in a revised form after attention is given to several issues.

Essential revisions:

Subsection “Lowering CDK activity reveals a cell cycle function for the Rim15-Igo1,2 pathway”, last paragraph: The authors argue that overexpression of Cln3 suppressed the large size and the longer G1 of the *cdc28-as1* cells grown without 1-NM-PP1. The authors should clarify this claim. Figure 1 indicates that G1 phase of the *cdc28-as1* allele is not longer that in wild type cells in SCRaf+Gal medium. Rather, G1 seems accelerated in cells expressing CLN3 and mitosis appears to take longer. Furthermore, it appears that the combination of *cdc28-as1* and *igo1,2* delta makes G1 faster (and mitosis longer). Is this consistent with the authors' main hypothesis?

Is the median cell size measured specifically for the G1 population? From examining Figure 1, it appears that the largest delay deriving from expression of the *cdc28-as1* allele takes place in mitosis, and that the size of cells correlates with the duration of mitosis (e.g. the smallest size is observed in cells with the fastest mitosis, the GAL-CLN3 (OFF) cells.

Introduction, third paragraph: the authors introduce the idea that the substrate phosphorylation status depends on the relative impact of both the kinase and phosphatase and cite a review article by Fisher et al. 2012. However, this concept was introduced in 2011, Phil. Trans. R. Soc. B (2011) 366, 3572-3583, which should be cited.

Introduction: 'Gwl does more than simply regulating mitosis in mammals (Charrasse et al. in revision)'. Can the authors be more specific here, especially as this paper is not yet published?

Subsection “Lowering CDK activity reveals a cell cycle function for the Rim15-Igo1,2 pathway”, last paragraph: '… cells were much larger… (98.5fL,…)…', as this is the first time that the cell volume measurements are introduced, it might be worth it to explain that median cell volumes are reported. The correct SI abbreviation for femtolitre is 'fl'.

Subsection “Lowering CDK activity reveals a cell cycle function for the Rim15-Igo1,2 pathway”, last paragraph: 'Deletion of IGO1 and IGO2 further increased the median size…, which means that Rim15-Igo1,2 is involved in cell size homeostasis'. In this and several other cases throughout the manuscript, the authors should be careful with the use of the terms 'size homeostasis' and 'size control'. Deletion of IGO1,2 leads to larger cells that still maintain a constant, albeit larger, size. In this case, it can be concluded that IGO1,2 contribute to cell size control, but not to cell size homeostasis. Only if deletion of IGO1,2 results in a larger spread of the cell size distribution, relative to the mean, one could conclude on a role in size homeostasis.

The data are generally well presented and clear. In some gels the level of loading standards seem to vary substantially from lane to lane, an aspect that could be improved.

Subsection “The Rim15 pathway is hyperactive in early G1 cells and repressed by Cln3”, first paragraph: 'No shift was detected in the igo1-S64A mutant, indicating that phosphorylation is due to Rim15.' It is not clear how this experiment shows that phosphorylation is due to Rim15. To conclude this, Igo1 phosphorylation should be compared in a rim15 deletion strain.

Subsection “The Rim15 pathway is hyperactive in early G1 cells and repressed by Cln3”, last paragraph: '…Igo1-S64 phosphorylation… decreased 2-fold within 10 min of Gal re-addition'. This is surprising as the cells were shifted to glucose-containing medium. Galactose induction is usually slow and sluggish once cells have been repressed with glucose. Has the experiment rather been performed by shift from Raff+Gal to Raff, then back to Raff+Gal?

Subsection “The Rim15-Igo1,2 pathway promotes START by inhibiting PP2A^Cdc55^”, last paragraph: '…expression of CLN2 from the MET3 promoter fully rescued the START delay of *cdc28-as1 igo1,2Δ* cells, indicating that the Rim15-Igo1,2 pathway favours the transcription of CLN1,2 genes.' This conclusion is not the only possible scenario. Increased PP2A activity in *igo1,2Δ* cells might simply pose a greater requirement for Cln-Cdk activity to overcome PP2A to trigger start. While increased CLN2 expression is one way to override PP2A, this does not necessarily mean that Igo1,2 impacts on CLN1,2 transcription. If the authors want to conclude on CLN1,2 transcription, CLN1,2 transcription should be experimentally assessed.

Subsection “Rim15/Gwl disrupts the balance between kinases and phosphatases”, first paragraph: 'Cell lacking Igo1,2 are larger while keeping the same doubling time…', this disagrees with the data presented in Figure 1, showing that the cell size of Igo1,2 lacking cells (62.3 fl) is very close to wt (61.6 fl).

Stb1 is apparently a cofactor of Whi5, and the authors should clearly explain this relationship. Of greater concern, the authors sometimes seem to say or imply that this protein is also a target of PP2A-Cdc55, but they have never looked at this protein on Western blots. The authors' approach to this cofactor needs to be made less confusing.

For the data in this paper to be convincing, it is essential that roughly 2X changes in Igo1 phosphorylation have consequence. Because of the presumed very tight association between the inhibitor and phosphatase, this should only be true if the concentration of PP2A-Cdc55 is in excess of the amount of phosphorylated Igo1,2 (see *eLife*. 2014 Mar 11;3:e01695). Shouldn't one in theory be able to mimic these results in diploid cells simply by comparing wild type with PP2A-Cdc55 heterozygotes? Here is a case where quantitation would clearly help make the authors' case stronger.

In Western blots, Swi6 is apparently a loading control, but then is the normalization across a blot measuring cell number or cell volume or…? For example, in Figure 1, why do the signals for Swi6 and Igo increase with time of incubation in NMPP1? This is never explained. How can the authors quantitate the 90 min timepoint on the Phostag gel when there is no good resolution between phosphorylated and unphosphorylated Igo1? The bar graph in Figure 1 is misleading because the y-axis does not show the full range; thus a 2-fold difference appears to be more significant than it actually is.

The experiments concerning chromosome loss shown in Figure 5 are interesting and suggestive, but the effects of the Rim15/Igo1,2 pathway on chromosome maintenance could be due to other substrates of PP2A-Cdc55 other than Whi5 and these effects could be independent of cell size. Experiments could be conceived to address this point; in any case, the authors rather blithely assume a connection that their experiments do not show.

Figure 6 can be improved: Why is the arrow from SBF to Cln1,2-Cdk black and not green? Whi-5P is a critical player, but it is shown in Gray and is almost invisible. Also, this figure tries to do too much and becomes somewhat confusing. I would prefer a figure that shows only what is happening when cells are grown in poor nutrient conditions, as this is the situation the work is intended to clarify. If needed, a related supplementary figure could show the situation in rich medium.

The reviewers also identified some issues with the presentation of the data, and in particular:

Introduction:

May be shortened, for instance by eliminating the discussion of models that have been already excluded and by reducing the discussion of G2-to-M transition in multicellular eukaryotes off. Furthermore, for readers who do not specialise in yeast, it would be good in the Introduction to better define the events of G1 (what is going on early and late) and explain the 3 types of G1 cyclins.

Results:

Subsection “Lowering CDK activity reveals a cell cycle function for the Rim15-Igo1,2 pathway”, second paragraph describing the experiments shown in Figure 1—figure supplement 1 are difficult to understand, particularly with respect to SWE1 deletion. The motivation for these experiments should be better explained.

Something is peculiar about the sentence “Most size regulation operating in G1 in budding yeast, our data suggest that the Rim15 pathway has a cryptic role in G1 control”.

Similarly, in the second paragraph of the subsection “The Rim15-Igo1,2 pathway promotes START by inhibiting PP2A^Cdc55^”, the references about suppressing pleiotropic effects of CDC55 deletion by deleting SWE1 are also obscure and are not easily interpretable by non-specialists.

Discussion:

Overall, the manuscript would benefit from a shortening of the Discussion, in particular by avoiding re-presenting material already presented in Results. Individual paragraphs are long and could be shortened to engage the reader.

The first paragraph of the subsection “Rim15/Gwl disrupts the balance between kinases and phosphatases” describes an analogy to Sin3-Rpd3 that is difficult to follow and quite speculative because apparently no evidence exists that Rim15 controls Ume6 via PP2A-B55. This is interesting but perhaps unnecessary; in any case the presentation could be improved.

The first paragraph of the subsection “Rim15, a nutrient-controlled starter for START in budding yeast” says that the system described in the paper operates when cells are grown in rich medium in conditions of low CDK activity. It seems very unlikely that wildtype yeast ever encounter such a situation; at least it should be acknowledged that this is an artificial setup.

One of the most interesting facets of this manuscript is the idea that CDKs regulate Rim15 in the opposite fashion that CDKs regulate Gwl in multicellular eukaryotes. In particular, the authors suggest that CDK phosphorylation excludes Rim15 from the nucleus. Given its importance, I would like to see a slight expansion of this topic. What did the three papers cited show? Are those results completely in line with the hypothesis? Where are all the players located during G1?

In the second paragraph of the subsection “Size homeostasis by negative feedback of Cln3-Cdk1 on Rim15”, a great deal of attention is given to the idea that TORC1 activity should decline at START. It is not clear if this is just a speculation or if there is any real experimental support to this idea. Given the author's success in isolating cells at various times in G1, why couldn't they simply measure this directly? And if this evidence does not exist, spending a half page on the topic seems unnecessary.

The last paragraph of the subsection “Size homeostasis by negative feedback of Cln3-Cdk1 on Rim15” talks about future experiments involving Rim15 overexpression possibly driving cells into passing START. It seems strange to introduce this idea; if those experiments are so important, why weren't they already done and included in this manuscript?

At least according to Figure 6 major conclusion of the manuscript is that Whi5 is a direct substrate of PP2A-Cdc55. Yet no attempt was made to purify the phosphatase and to get a general idea of whether the presumptive reaction actually occurs and if so whether its kinetic parameters support the hypothesis. The authors should point this out.

If possible, the authors should briefly discuss these questions: Why is it advantageous for yeast cells to divide when they are smaller under limiting nutrient conditions? Why wouldn't it be better to delay division and go into some kind of spore-like state? Conversely, why would it not be advantageous for cells to divide earlier when grown in rich medium? I realize this could easily reach into the realm of speculation, but nonetheless the underlying logic of the phenomenology is not without importance.

---

## [Author Response]

*Essential revisions:*

*Subsection “Lowering CDK activity reveals a cell cycle function for the Rim15-Igo1,2 pathway”, last paragraph: The authors argue that overexpression of Cln3 suppressed the large size and the longer G1 of the cdc28-as1 cells grown without 1-NM-PP1. The authors should clarify this claim. Figure 1 indicates that G1 phase of the cdc28-as1 allele is not longer that in wild type cells in SCRaf+Gal medium. Rather, G1 seems accelerated in cells expressing CLN3 and mitosis appears to take longer. Furthermore, it appears that the combination of cdc28-as1 and igo1,2 delta makes G1 faster (and mitosis longer). Is this consistent with the authors' main hypothesis?*

We repeated the experiments shown in Figure 1 with *GAL-CLN3* strains grown for a longer time in SCD or in SCRaf+Gal media. These experiments are done on asynchronous cell populations and G1 duration is calculated by multiplying the fraction of 1C DNA cells (measured from EdU/PI bivariate FACS) by the population doubling time. Thus *cdc28-as1* cells, which have a START delay, will not spend proportionally more time in G1 compared to WT cells because they are also born at a larger size. In contrast, if G1 duration is measured from small elutriated cells it will be longer for *cdc28-as1* then for WT cells (see Figure 3). The same logic applies when comparing *cdc28-as1* and *cdc28-as1 igo1,2∆* cells: the latter appear to spend less time in G1 in this population-based assay, mainly because they are born larger. Our data clearly show that Cln3 overexpression lowers the mean size of *cdc28-as1* cells (60.8 fL vs. 74.9 fL; Figure 1) and the fraction of G1 cells in the population (Figure 1—figure supplement 1). These data also indicate, as noticed by the reviewer, that WT cells overexpressing Cln3 compensate their shorter G1 by spending more time in G2+M; we don’t know whether this reflects a cryptic size control in G2 or whether subtle defects in cell cycle progression activate checkpoint responses. These notions are now clarified in the manuscript.

*Is the median cell size measured specifically for the G1 population? From examining Figure 1, it appears that the largest delay deriving from expression of the cdc28-as1 allele takes place in mitosis, and that the size of cells correlates with the duration of mitosis (e.g. the smallest size is observed in cells with the fastest mitosis, the GAL-CLN3 (OFF) cells.*

The median cell size is for the entire population, not solely for G1 cells. It is believed that *cdc28-as1* cells are delayed both at the G1/S and G2/M transitions, even in the absence of inhibitor, due to lower catalytic activity. It is true that the G2+M delay of *cdc28-as1* cells seems larger than that in G1, but caution is required since cells born larger will necessarily spend less time in G1 before reaching the critical cell size for START. Addition of 1-NM-PP1 mostly slows down mitosis (Figure 1—figure supplement 1), as expected from the higher CDK activity required for mitosis. Interestingly S phase duration remains relatively constant (~30 min) in all strains tested. Data with *GAL-CLN3* (OFF) cells has been corrected, being now larger compared to WT, as expected.

*Introduction, third paragraph: the authors introduce the idea that the substrate phosphorylation status depends on the relative impact of both the kinase and phosphatase and cite a review article by Fisher et al. 2012. However, this concept was introduced in 2011, Phil. Trans. R. Soc. B (2011) 366, 3572-3583, which should be cited.*

Thank you for the suggestion. We corrected this omission.

*Introduction: 'Gwl does more than simply regulating mitosis in mammals (Charrasse et al. in revision)'. Can the authors be more specific here, especially as this paper is not yet published?*

This paper, now in press (Nature Commun), shows that ENSA depletion in human cells leads to severe S-phase extension through down regulation of Treslin/TICRR (a CDK target essential for the firing of replication origins), indicating that Gwl also plays roles for DNA replication.

*Subsection “Lowering CDK activity reveals a cell cycle function for the Rim15-Igo1,2 pathway”, last paragraph: '… cells were much larger… (98.5fL,…)…', as this is the first time that the cell volume measurements are introduced, it might be worth it to explain that median cell volumes are reported. The correct SI abbreviation for femtolitre is 'fl'.*

We now indicate that median cell volume is used to characterize asynchronous cell population. The internationally recommended abbreviation for liter is now L (not l) and thus femtoliter should be written fL (https://en.wikipedia.org/wiki/Litre; http://www.convertunits.com/info/femtoliter)

*Subsection “Lowering CDK activity reveals a cell cycle function for the Rim15-Igo1,2 pathway”, last paragraph: 'Deletion of IGO1 and IGO2 further increased the median size…, which means that Rim15-Igo1,2 is involved in cell size homeostasis'. In this and several other cases throughout the manuscript, the authors should be careful with the use of the terms 'size homeostasis' and 'size control'. Deletion of IGO1,2 leads to larger cells that still maintain a constant, albeit larger, size. In this case, it can be concluded that IGO1,2 contribute to cell size control, but not to cell size homeostasis. Only if deletion of IGO1,2 results in a larger spread of the cell size distribution, relative to the mean, one could conclude on a role in size homeostasis.*

Thank you for bringing up this relevant point. We now provide in Figure 1—figure supplement 2 a profile of cell size distributions for asynchronous populations of *cdc28-as1* and *cdc28-as1 igo1,2∆* cells grown in SC-D (left panel), where it can be seen that the latter strain has a larger mean but also wider spread of size distribution. We use the term “cell size homeostasis” also because we identified feedback mechanisms that buffer excessive cell size fluctuations by inactivating Rim15 (increasing PP2A activity on Whi5) when Cln3 is too high, and conversely. We agree however that “cell size control” and “cell size homeostasis” should be used more cautiously, something we tried to do in the revised manuscript.

*The data are generally well presented and clear. In some gels the level of loading standards seem to vary substantially from lane to lane, an aspect that could be improved.*

We agree that protein loading is sometimes variable between lanes, but all quantifications are presented as ratio of protein over loading control or of phospho- and unphosphorylated bands to take into account these variations.

*Subsection “The Rim15 pathway is hyperactive in early G1 cells and repressed by Cln3”, first paragraph: 'No shift was detected in the igo1-S64A mutant, indicating that phosphorylation is due to Rim15.' It is not clear how this experiment shows that phosphorylation is due to Rim15. To conclude this, Igo1 phosphorylation should be compared in a rim15 deletion strain.*

Targeting of Igo1-S64 by Rim15 and lack of Ser64 phosphorylation in *rim15*∆ cells has been demonstrated by the De Virgilio lab (Talarek et al., 2010; Bontron et al., 2013). The peptide phosphorylated is highly conserved in Igo1 orthologs from other species (Juanes et al., 2013) where direct phosphorylation by Rim15/Gwl has been shown (Chica et al., 2016). This is now clarified in the text.

*Subsection “The Rim15 pathway is hyperactive in early G1 cells and repressed by Cln3”, last paragraph: '…Igo1-S64 phosphorylation… decreased 2-fold within 10 min of Gal re-addition'. This is surprising as the cells were shifted to glucose-containing medium. Galactose induction is usually slow and sluggish once cells have been repressed with glucose. Has the experiment rather been performed by shift from Raff+Gal to Raff, then back to Raff+Gal?*

No, cells were shifted from Raff+Glu to Raff+Gal, but using filtration and extensive washing with SC-Raff+Gal to remove all glucose from the medium. This allows for fast activation of the GAL promoter.

*Subsection “The Rim15-Igo1,2 pathway promotes START by inhibiting PP2A^Cdc55^”, last paragraph: '…expression of CLN2 from the MET3 promoter fully rescued the START delay of cdc28-as1 igo1,2Δ cells, indicating that the Rim15-Igo1,2 pathway favours the transcription of CLN1,2 genes.' This conclusion is not the only possible scenario. Increased PP2A activity in igo1,2Δ cells might simply pose a greater requirement for Cln-Cdk activity to overcome PP2A to trigger start. While increased CLN2 expression is one way to override PP2A, this does not necessarily mean that Igo1,2 impacts on CLN1,2 transcription. If the authors want to conclude on CLN1,2 transcription, CLN1,2 transcription should be experimentally assessed.*

We agree that we have not shown directly that Rim15 activity favors *CLN1,2* transcription, yet the suppression of *cdc28-as1 igo1,2∆*’s larger size at START by *whi5∆ stb1∆* double deletion (Figure 4) really suggests this might be the case. We have now toned down this point in the Results section to discuss it later.

*Subsection “Rim15/Gwl disrupts the balance between kinases and phosphatases”, first paragraph: 'Cell lacking Igo1,2 are larger while keeping the same doubling time…', this disagrees with the data presented in Figure 1, showing that the cell size of Igo1,2 lacking cells (62.3 fl) is very close to wt (61.6 fl).*

We meant to say that *cdc28-as1* cells lacking Igo1,2 are larger. This is now clarified in the manuscript.

*Stb1 is apparently a cofactor of Whi5, and the authors should clearly explain this relationship. Of greater concern, the authors sometimes seem to say or imply that this protein is also a target of PP2A-Cdc55, but they have never looked at this protein on Western blots. The authors' approach to this cofactor needs to be made less confusing.*

We have now better explained the relationship between Whi5 and Stb1 in the text. We had performed Phos-tag analysis of Stb1 in *cdc28-as1* cells lacking or not Igo1,2, which indicates that Stb1 phosphorylation is also delayed in *igo1,2∆* cells (occurring at 73 fL instead of 57 fL). The blot, unfortunately not of “publication quality”, is provided as Figure 7. We now refer to this experiment in the manuscript as data not shown.

Author response image 1.**DOI:**
http://dx.doi.org/10.7554/eLife.26233.017

*For the data in this paper to be convincing, it is essential that roughly 2X changes in Igo1 phosphorylation have consequence. Because of the presumed very tight association between the inhibitor and phosphatase, this should only be true if the concentration of PP2A-Cdc55 is in excess of the amount of phosphorylated Igo1,2 (see eLife. 2014 Mar 11;3:e01695). Shouldn't one in theory be able to mimic these results in diploid cells simply by comparing wild type with PP2A-Cdc55 heterozygotes? Here is a case where quantitation would clearly help make the authors' case stronger.*

According to Ghaemmaghami et al. (2003) and the yeast database (SGD), there is ~2x more Cdc55 than Igo1,2 proteins, which would allow a two-fold change in Igo phosphorylation to have consequences. Monitoring START in diploids heterozygous for *CDC55* might be a way to test this. However subcellular localization, which is tightly regulated for Cdc55 and Rim15 should also be taken into account (Rossio et al., 2011; Wanke et al. 2005; Moreno-Torres et al., 2017). Thus, it is possible that nuclear Igo1,2 phosphorylation is reduced more than 2-fold in the nucleus at START.

*In Western blots, Swi6 is apparently a loading control, but then is the normalization across a blot measuring cell number or cell volume or…? For example, in Figure 1, why do the signals for Swi6 and Igo increase with time of incubation in NMPP1? This is never explained. How can the authors quantitate the 90 min timepoint on the Phostag gel when there is no good resolution between phosphorylated and unphosphorylated Igo1? The bar graph in Figure 1 is misleading because the y-axis does not show the full range; thus a 2-fold difference appears to be more significant than it actually is.*

Normalization in western blots is done according to total protein amount (10 µg/lane). However, Bradford protein quantification can be inaccurate with the TCA extracts we use, explaining the differences between lanes. This is why our quantifications are always normalized to a loading control (e.g. Swi6, which is constant during the cell cycle), or directly as the% of phospho-Igo1 (phosphorylated/total Igo1). Because the 10 nM 1-NM-PP1 (90 min) point is hard to quantitate we added lanes with 50 nM 1-NM-PP1, where the 90 min time point is quantifiable. The Y-scale starts at 25% because it is more or less the level of background signal seen with the S64A non-phosphorylable mutant. This figure shows that the fraction of phospho-Igo1 increases ≥2-fold quickly following down regulation of Cdc28 activity with low amounts of a specific inhibitor.

*The experiments concerning chromosome loss shown in Figure 5 are interesting and suggestive, but the effects of the Rim15/Igo1,2 pathway on chromosome maintenance could be due to other substrates of PP2A-Cdc55 other than Whi5 and these effects could be independent of cell size. Experiments could be conceived to address this point; in any case, the authors rather blithely assume a connection that their experiments do not show.*

We fully agree. We have no direct evidence that the increased chromosome loss seen in *cdc28-as1 igo1,2∆* cells, or in *igo1,2∆* cells grown on glycerol/lactate, depends on Whi5. The fact that other PP2A^Cdc55^ substrates might be involved is now stated. However, we present new data showing that Rim15 overexpression also increases chromosome loss in *cdc28-as1* cells (Figure 5) while decreasing cell size (Figure 1—figure supplement 2). Knowing that size control mainly operates at START through Cln3 and Whi5 in yeast and that *whi5∆* mutants also lose chromosome (data not shown; Pan et al. 2006), it is tantalizing to speculate that deregulation of the Rim15-Igo1,2 pathway affects chromosome maintenance through the Whi5/START module. This is now stated as such in the text.

*Figure 6 can be improved: Why is the arrow from SBF to Cln1,2-Cdk black and not green? Whi-5P is a critical player, but it is shown in Gray and is almost invisible. Also, this figure tries to do too much and becomes somewhat confusing. I would prefer a figure that shows only what is happening when cells are grown in poor nutrient conditions, as this is the situation the work is intended to clarify. If needed, a related supplementary figure could show the situation in rich medium.*

Thank you for the suggestion that we find very wise. The schematic is now redrawn to emphasize the situation in poor nutrient conditions, when TORC1 is low, Rim15 high and PP2A down.

*The reviewers also identified some issues with the presentation of the data, and in particular:*

*Introduction:*

*May be shortened, for instance by eliminating the discussion of models that have been already excluded and by reducing the discussion of G2-to-M transition in multicellular eukaryotes off. Furthermore, for readers who do not specialise in yeast, it would be good in the Introduction to better define the events of G1 (what is going on early and late) and explain the 3 types of G1 cyclins.*

The Introduction has been shortened and the role of the three G1 cyclins better explained.

*Results:*

*Subsection “Lowering CDK activity reveals a cell cycle function for the Rim15-Igo1,2 pathway”, second paragraph describing the experiments shown in Figure 1—figure supplement 1 are difficult to understand, particularly with respect to SWE1 deletion. The motivation for these experiments should be better explained.*

The motivation for the experiments with *SWE1* deletion is now better explained

*Discussion:*

*Overall, the manuscript would benefit from a shortening of the Discussion, in particular by avoiding re-presenting material already presented in Results. Individual paragraphs are long and could be shortened to engage the reader.*

We agree. The Discussion is now shorter.

*The first paragraph of the subsection “Rim15/Gwl disrupts the balance between kinases and phosphatases” describes an analogy to Sin3-Rpd3 that is difficult to follow and quite speculative because apparently no evidence exists that Rim15 controls Ume6 via PP2A-B55. This is interesting but perhaps unnecessary; in any case the presentation could be improved.*

We agree. This analogy has been removed.

*The first paragraph of the subsection “Rim15, a nutrient-controlled starter for START in budding yeast” says that the system described in the paper operates when cells are grown in rich medium in conditions of low CDK activity. It seems very unlikely that wildtype yeast ever encounter such a situation; at least it should be acknowledged that this is an artificial setup.*

The *cdc28-as1* strain with low Cdk activity is clearly an artificial system that helped uncover a function for the Rim15-Igo1,2 pathway. This is acknowledged in the text. However, it could be argued that growing cells in rich laboratory conditions is also artificial from a yeast standpoint. Crucially, we show that the Rim15 pathway regulates START in wild type yeast growing by oxidative phosphorylation (respiration), a condition that is highly relevant for the physiology of yeast and higher organisms.

*One of the most interesting facets of this manuscript is the idea that CDKs regulate Rim15 in the opposite fashion that CDKs regulate Gwl in multicellular eukaryotes. In particular, the authors suggest that CDK phosphorylation excludes Rim15 from the nucleus. Given its importance, I would like to see a slight expansion of this topic. What did the three papers cited show? Are those results completely in line with the hypothesis? Where are all the players located during G1?*

This topic is now expanded and information on the cited references included.

*In the second paragraph of the subsection “Size homeostasis by negative feedback of Cln3-Cdk1 on Rim15”, a great deal of attention is given to the idea that TORC1 activity should decline at START. It is not clear if this is just a speculation or if there is any real experimental support to this idea. Given the author's success in isolating cells at various times in G1, why couldn't they simply measure this directly? And if this evidence does not exist, spending a half page on the topic seems unnecessary.*

We invested significant effort during the revision of this paper to set up an assay monitoring TORC1 activity throughout G1 of elutriated cells. Using Rps6 (ribosomal S6 protein) phosphorylation as a proxy for TORC1 activity, we obtained some evidence for TORC1 activity rising during G1 (Figure 2—figure supplement 2), which would be consistent with the concomitant decline in Rim15 activity. Unfortunately, our antibodies against total Rps6 were not good enough to detect Rps6 in early G1 cells, making difficult to correctly quantitate TORC1 activity. Since we cannot ascertain that TORC1 declines at START due to the lower surface-to-volume ratio of cells, we decided to fully remove this speculation from the manuscript, and leave it for future investigations.

*The last paragraph of the subsection “Size homeostasis by negative feedback of Cln3-Cdk1 on Rim15” talks about future experiments involving Rim15 overexpression possibly driving cells into passing START. It seems strange to introduce this idea; if those experiments are so important, why weren't they already done and included in this manuscript?*

We now include novel data on Rim15 overexpression showing that it decreases cell size in wild type, *cdc28-as1* and *cln3∆* strains, and triggers chromosome loss in *cdc28-as1* cells. These data reinforce the notion that the Rim15-Igo1,2 pathway controls cell size in yeast, with hyper-activation or ablation leading to smaller or larger cell size, respectively. This is a strong indication that Rim15 overexpression drives cells into passing START in an unscheduled way.

*At least according to Figure 6 major conclusion of the manuscript is that Whi5 is a direct substrate of PP2A-Cdc55. Yet no attempt was made to purify the phosphatase and to get a general idea of whether the presumptive reaction actually occurs and if so whether its kinetic parameters support the hypothesis. The authors should point this out.*

We agree with this point and now state that no attempts were made, at this stage, to biochemically test our model. This goes beyond the scope of this work and is left for future investigations.

*If possible, the authors should briefly discuss these questions: Why is it advantageous for yeast cells to divide when they are smaller under limiting nutrient conditions? Why wouldn't it be better to delay division and go into some kind of spore-like state? Conversely, why would it not be advantageous for cells to divide earlier when grown in rich medium? I realize this could easily reach into the realm of speculation, but nonetheless the underlying logic of the phenomenology is not without importance.*

One possible advantage for cells to adopt a smaller size when grown on poor nutrients is to optimize nutrients uptake relative to cell volume. Indeed, the surface area-to-volume ratio of cells decreases logarithmically when cells increase in size. With nutrient transporters embedded in the plasma membrane, smaller cells would be able to import more nutrients relative to their small volume. These notions are developed in Ginzberg & Kirschner (2015) and now mentioned in the Discussion.